# Review on the Chemistry of [M(NH₃)ₙ](XO₄)ₘ (M = Transition Metal, X = Mn, Tc or Re, n = 1–6, m = 1–3) Ammine Complexes

**Raj Narain Mehrotra**

Chemistry, Chemistry Department, JNV University, Jodhpur (Raj) 342005, India; rnmehrotra54@gmail.com

**Abstract:** The preparation of ammine complexes of transition metals having oxidizing anions such as permanganate and perrhenate ions is a great challenge due to possible reactions between ammonia and oxidizing anions during the synthesis of these materials. However, it has an important role in both the development of new oxidants in organic chemistry and especially in the preparation of mixed-metal oxide catalyst precursors and metal alloys for their controlled temperature decomposition reactions. Therefore, in this paper, synthetic procedures to prepare ammonia complexes of transition metal permanganate, pertechnetate, and perrhenate (the VIIB group tetraoxometallates) salts have been comprehensively reviewed. The available data about these compounds' structures and spectroscopic properties, including the presence of hydrogen bonds that act as redox reaction centers during thermal decomposition, are given and evaluated in detail. The nature of the thermal decomposition products has also been summarized. The available information about the role of the ammine complexes of transition metal permanganate salts in organic oxidation reactions, such as the oxidation of benzyl alcohols and regeneration of oxo-compounds from oximes and phenylhydrazones, including the kinetics of these processes, has also been collected. Their physical and chemical properties, including the thermal decomposition characteristics of known diammine (Ag(I), Cd, Zn, Cu(II), Ni(II)), triammine (Ag(I)), and simple or mixed ligand tetraammine (Cu(II), Zn, Cd, Ni(II), Co(II), Pt(II), Pd(II), Co(III)), Ru(III), pentaammine (Co(III), Cr(III), Rh(III) and Ir(III)), and hexaammine (Ni(II), Co(III), Cr(III)) complexes of transition metals with tetraoxometallate(VII) anions (M = Mn, Tc and Re), have been summarized. The preparation and properties of some special mixed ligand/anion/cation-containing complexes, such as [Ru(NH₃)₄(NO)(H₂O)](ReO₄)₂, [Co(NH₃)₅(H₂O)](ReO₄)₂, [Co(NH₃)₅X](MnO₄)₂ (X = Cl, Br), [Co(NH₃)₆]Cl₂(MnO₄), [Co(NH₃)₅ReO₄]X₂ (X = Cl, NO₃, ClO₄, ReO₄), and K[Co(NH₃)₆]Cl₂(MnO₄)₂, are also included.

**Keywords:** ammonia; ammine; crystal structure; synthesis; spectroscopy; hydrogen bonds; oxidation; thermal decomposition; spinels; permanganate; pertechnetate; perrhenate

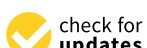



## 1. Introduction

The temperature-controlled decomposition of transition metal complexes containing redox-active central atoms, ligands, and anions in solid-phase quasi-intramolecular redox reactions has become a widely used method of preparing nano-sized oxides, nitrides, carbides, metals, and their alloys [1–19]. The reducing ligands (pyridine, urea, or ammonia) and oxidizing anions containing precursors frequently give amorphous decomposition intermediates, which can be transformed into controlled-sized crystalline materials by heat treatment [7,10–28]. Permanganate complexes and other tetraoxometallates result in the formation of various nano-sized mixed oxides, especially MᴬMᴮ₂O₄ spinels, including those with varying ratios of Mᴬ and Mᴮ cations at the tetrahedral or octahedral sites of the spinel lattice (Mᴬ being the cationic and Mᴮ the anionic metal component) [29]. There are many possible ways to adjust the ratio of Mᴬ and Mᴮ metals in the formed oxides/spinels. For example, one way is through the variation of Mᴬ and Mᴮ cation ratios, as in the case of [Co(NH₃)₆](MnO₄)₃ (Co:Mn = 1:3), [Co(NH₃)₅Cl](MnO₄)₂ (Co:Mn = 1:2),

or $[Co(NH_3)_6]Cl_2MnO_4$ and $[Co(NH_3)_4CO_3]MnO_4$ (Co:Mn = 1:1) using inner- or outer-sphere co-anions, which decomposes into gases. Another possibility is the use of $(M^{A1}, M^{A2} \ldots ..)((M^{B1}O_4, M^{B2}O_4, \ldots ..)_n$ solid solutions, change in the valence of transition metals (e.g., $Co^{II}/Co^{III}$, $Fe^{II}/Fe^{III}$), or the charge or polymerization degree of the oxidizing anions ($MnO_4^-/MnO_4^{2-}$, $CrO_4^{2-}/Cr_2O_7^{2-}$). Isomorph solid solutions with metal-free tetraoxo-nonmetallate anions (e.g., perchlorate, sulfate), which decompose into gaseous products [30–33], ensure the preparation of mixed crystals with partial substitution of the metal-containing anions with metal-free ones [34]. Two monovalent anion-containing compounds may be isomorphic with compounds containing a divalent cation and a neutral component; e.g., $[M(NH_3)_4](XO_4)_2$ (M = Zn, Cd, X = Cl, Mn) complexes are isomorphic with $[Zn(NH_3)_4]MoO_4 \cdot H_2O$ [35–37].

The redox activity of each component (cations, anions, and ligands) in these complexes, however, does not follow the redox potential order found for these species in aqueous solutions. The special oxidation properties of these complexes have enormous importance in organic synthesis as mild oxidants in organic media [11,14,38–49] and strong oxidants in acidic aqueous environments [50–54].

One of the most promising groups of the above-mentioned complexes is the ammonia complex group [34,55–58], especially the complexes with tetraoxometallate anions of the VIIB group (Mn, Tc, Re). These complexes show wide variation in coordination number and geometry, which has an influence on the temperature-initiated redox reactions in these complexes, found not only between the anion and ligand [28,29,37,54,59,60] but in some cases between the oxidizing central ion (e.g., cobalt(III)) and the ammonia as well [61,62]. This in situ ammonia formation, e.g., from urea complexes [11], shows reactions similar to the decomposition of ammonia, with ammonia being released together with the oxidation reaction. Therefore, in this review, the preparation, properties, thermal decomposition, and organic oxidation reactions of complexes formed from the transition metal tetraoxometallate ($XO_4^-$, X = Mn, Tc, and Re) and ammonia are reviewed.

## 2. Discussion

### 2.1. General Consideration on the Synthesis of Ammonia Complexed Transition Metal Tetraoxometallates ($XO_4^-$, X = Mn, Tc, Re)

In the preparation of compounds having redox-active components, namely, transition metal complexes with ammonia ligands and oxidizing anions such as permanganate ions, the main challenge is the preparation of the title compounds without the oxidation of the ligand with the anion during synthesis. Some of the easy synthesis routes, e.g., reactions of manganese heptoxide, permanganic acid, or aluminum permanganate with basic precursors, may not be suitable for ammonia complexes due to the strong oxidation ability of $Mn_2O_7$ or permanganate reactants [63–65]. Metathesis reactions using barium permanganate and sulfate compounds [64,66] are not advantageous because the ammonia complexes of transition metal permanganates generally do not dissolve well in water, and only diluted solutions can be prepared when the formed salt is at a neutral or uncontrolled pH. The metathesis reactions of readily soluble transition metal ammine complexes and water-soluble permanganates such as sodium or potassium permanganate, however, ensure the chance for easy control of the oxidation potential of permanganate ion in the aqueous solutions used during synthesis. The oxidation ability of pertechnetate and perrhenate ions under similar conditions is much lower than that of permanganate ions; therefore, the conditions applicable to the synthesis of permanganates may be expected to be useful for the synthesis of pertechnetate or perrhenate complexes. Therefore, our discussion is limited to permanganate ions. The Nernst equation can be written as

$$E = E_0 - (RT/nF)ln([RED]/[OX]),$$

which unambiguously shows that the concentration of the oxidant and the temperature are key factors in these reactions. The pH controls the number of electrons ($n$) in the permanganate oxidations; namely, a permanganate oxidizes with five (acidic medium),

three (neutral medium), or only one electron (basic medium). Thus, the pH control is one of the most important factors in these synthesis reactions. Therefore, the key factors in avoiding oxidation of ammonia ligand with permanganate (or other tetraoxometallates) anions during the synthesis of the transition metal ammine complex salts, especially if we use metathesis reactions between the water-soluble salts of transition metal ammine complexes and soluble permanganate salts (K, Na or Ba salts [65]), are the following:

1.  The low concentration of permanganate ion in the solution is advantageous and can easily reach if the permanganate complex prepared is sparingly soluble. In this case, the decreasing solubility with the salting out effect by using the excess of the most soluble salt (chloride, nitrate) of the complex cation and decreasing the temperature is advantageous. A starting salt in a metathesis reaction with permanganates (transition metal ammonia complex salt, e.g., chloride, nitrate, etc.) is to be selected among the most soluble salts and used as a concentrated solution as possible;

2.  The decrease in temperature diminishes the oxidation ability of permanganate ions and generally decreases the solubility, thus acting as a main driving force in the removal of the oxidizing anion from the solution. Therefore, the typical synthesis can be conducted at and below room temperature, with immediate cooling of the solution until freezing;

3.  The excess ammonia increases the pH and increases the concentration of the ammonia-complexed cations in the solution; thus, using the ammonia in excess is also a key step in these syntheses. It is the most important factor in decreasing the oxidation ability of permanganate ions.

Keeping these conditions, the majority of the known permanganate [65], pertechnetate, and perrhenate [12] salts of transition metal ammonia complexes can easily be prepared. The other possible reaction route is when solid transition metal tetraoxometallates are reacted with gaseous or liquid ammonia in the absence of water. The coordination number of transition metal toward ammonia, however, depends on the nature of the metal and the synthesis conditions, and in general, the solid transition metal salts with ammonia without solvent resulted in complexes of higher coordination number [67].

### 2.2. Diammine Complexes

Furthermore, only a few silver(I) diammine complexes of Zn, Cd, and Ni have been discovered so far. Two diamminesilver permanganates, the hydrated form of $[Ag(NH_3)_2]MnO_4$, the perrhenate complexes of Ag(I), Zn(II), Cd(II), and Ni(II) [68] have been described up until now. No diammine complexes of metal pertechnetates have been described up until now.

### 2.2.1. Preparation and Properties of Diamminesilver(I) Permanganate $[Ag(NH_3)_2]MnO_4$

The first complex permanganate salt prepared in chemical history was the diamminesilver(I) permanganate. It was isolated by Klobb [69] in the reaction of an ammoniacal solution of silver nitrate with potassium permanganate at 10 °C. In a similar way, Scagliari and Marangoni [70] prepared a compound described as monohydrate, $AgMnO_4 \cdot 2NH_3 \cdot H_2O$. Bruni and Levi repeated this experiment; however, they found that the anhydrous salt was formed [67]. The $[Ag(NH_3)_2]MnO_4$ explodes by percussion and gradually decomposes in air by losing ammonia. It forms violet rhombic plate-like crystals, sparingly soluble in cold and better soluble in warm water [69]. Its solubility is 3.6 g/100 mL of water at 20 °C [65]. However, in hot water, it easily hydrolyses [71,72] with ammonia evolution. Heating of the aqueous solution of [diamminesilver(I)] permanganate to remove the ammonia did not result in the expected $AgMnO_4$, instead an ammonium permanganate and silver(I) oxide precipitate was formed. The dissociation of the complex cation with ammonia formation resulted in protonation of the liberated ammonia with ammonium and hydroxide ion formation, and the latter forms insoluble $Ag_2O$, and ammonium ions are accumulated in the solution. During standing or heating of the solution, the permanganate ion concentration did not change; however, the cation concentration decreased [70,71].

The detailed study of Kótai et al. showed that half of the ammonia is evolved as gaseous ammonia, and the other part is protonated and left in the solution as ammonium ion [70,71,73].

$$2[Ag(NH_3)_2]MnO_4 + H_2O = 2NH_4MnO_4 + Ag_2O + 2NH_3$$

The removal of the excess ammonia from the equilibrium with heating or using a vacuum quickens the process, whereas by allowing the natural evaporation on standing at room temperature with a low ammonia vapor pressure above the solution, monocrystals of ammonium permanganate could be grown [73]. This reaction, because of the equilibrium between the dissociated–dissolved–evaporated ammonia, keeps a constant high pH due to the presence of liberated ammonia and stabilizes the ammonium permanganate solution; thus, a slow crystallization process could be carried out without oxidation of ammonia or ammonium ions in the solution. The other methods to prepare ammonium permanganate [65] resulted in ammonium permanganate in the reactions conducted at not alkaline conditions; thus, fast precipitation was used to remove that from the solution. The simplest preparation method used acidic ammonium chloride [74], leading to mixed $(K,NH_4)MnO_4$ [65], which, on repeated recrystallization, resulted in the enrichment of K-content [65,75,76].

Fogaca et al. studied the earlier published synthesis routes and found that all methods produced the same diamminesilver(I) permanganate without crystalline water [34]. The synthesis of pure material was performed with the reaction of $[Ag(NH_3)_2]NO_3$ and $NaMnO_4$ solutions at room temperature. The anhydrous $[Ag(NH_3)_2]MnO_4$ has two monoclinic polymorphs ($a$ = 7.9095 Å, $b$ = 6.0205 Å, $c$ = 12.6904 Å, $\beta$ = 98.056, $V$ = 598.34 Å$^3$, $d$ = 2.896 P2/m, and $a$ = 7.8112 Å, $b$ = 6.0682 Å, c = 13.1260 Å, $\beta$ = 96.4388, $V$ = 618.25 Å$^3$, $d$ = 2.803, I2/m) for the low- and high-temperature modifications, respectively. The phase transition temperature is 162.3 K, and the heat of transformation was found to be 1.107 kJ/mol.

Scagliari and Marangoni defined isomorphism between the diamminesilver(I) permanganate and perchlorate (both compounds were supposed to be monohydrate) because of the formation of purple mixed crystals with varying colors depending on the ratio of components [70]. The diamminesilver(I) perchlorate is, however, monoclinic at low temperatures ($T_{m-o}$ = 225.7 K) and has an orthorhombic form at room temperature. The low-temperature polymorphs of the permanganate and the perchlorate complex are isomorphous [30,34]; however, the room temperature polymorphs (monoclinic and orthorhombic) different, although the orthorhombic cell is a special case of monoclinic one, with $\beta$ = 90°. Since the cell volumes are close to each other [30,34], the formation of continuous solid solutions could be observed according to the following:

The reaction of $[Ag(NH_3)_2]NO_3$ with $(K,Na)(MnO_4,ClO_4)$-containing solutions with smaller than 3:7 $MnO_4^-/ClO_4^-$ molar ratio resulted in purple precipitates even at room temperature, whereas the solution with 1:1 $MnO_4^-/ClO_4^-$ molar ratio required cooling to obtain any crystalline product. The solid solution products with $MnO_4^-/ClO_4^-$ > 3 molar ratios (solution phase) could be prepared with the use of highly soluble NaMnO4 [65]. Increasing the $MnO_4^-/ClO_4^-$ molar ratio in the solutions resulted in a continuous increase of the $MnO_4^-/ClO_4^-$ ratio in the formed solid solutions. Monoclinic and orthorhombic solid solutions were isolated without a miscibility gap, and the phase transformation occurred with ~28 mol % permanganate content. The spectroscopic and XRD characterization of solid solutions are also described, but the thermal decomposition was only checked for the pure $[Ag(NH_3)_2]ClO_4$ [77], which shows that the reaction routes are very different; thus, the studies on the thermal decomposition of solid solutions are expected to give exciting new results.

The asymmetric unit of the low- and high-temperature modification contains four quarters of Ag, four halves of $NH_3$, two halves of permanganate, and two quarters of Ag, two halves of $NH_3$, and a half of permanganate, respectively, due to higher symmetry of the high-temperature modification. A unique 3D coordination network is built up in both poly-

morphs with the formation of chain-like Ag–Ag bonds, and the permanganate anions were coordinated with every second silver cation. The geometry around every second Ag cation is octahedral (2-2 neighboring silver (argentophilic) interactions), two permanganates, and two $NH_3$ molecules. The axial silver cations have SP-4 geometry (two octahedral silver ions via argentophilic interactions and two ammonia ligands) (Figure 1). The ammonia ligands, the permanganates, and the silver cations are in all-trans arrangements. The high-temperature modification contains more distorted coordination polyhedrons than the low-temperature one.

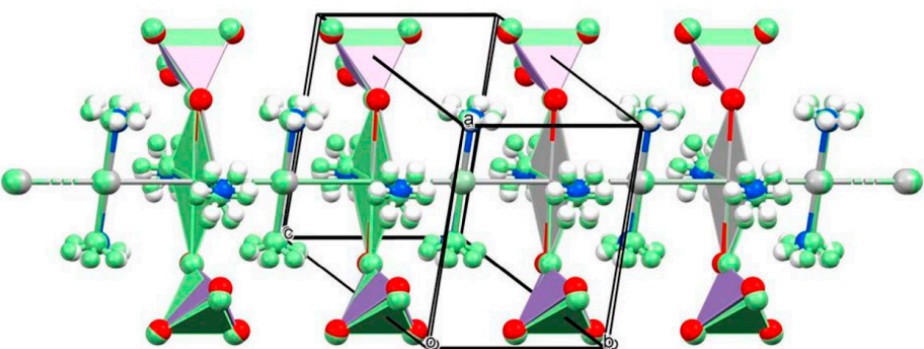

**Figure 1.** The crystal structure of $[Ag(NH_3)_2MnO_4]$. Reproduced from [34].

The coordinated ammonia molecules are disordered in both structures. The infinite ladder-like chains of silver ions are parallel to the *b*-axis and coincide with the 2-fold rotation axes in both structures. The Ag–Ag distances are half of the length of the *b*-axis, and the Ag ions sit on the inversion centers in each polymorph. A detailed structural evaluation has been given for both polymorphs [34]. Four and two crystallographically different permanganate ion environments were defined and evaluated in the correlation analysis of the low- and high-temperature polymorphs, respectively. Their IR and Raman bands were assigned completely [34]. The hydrogen bond parameters and their influence on the IR and Raman spectroscopical features were also discussed in detail [34].

Exothermic decomposition of the complex can be observed at 85 °C with the liberation of one mol of ammonia; then, a second decomposition step occurs at 204 °C. Water also forms in both steps, which shows that the ammonia and permanganate ions reacted with each other even in the first reaction step (the only sources of hydrogen and oxygen for water formation are ammonia and permanganate ions, respectively). The formal main decomposition step can be written as follows:

$$2[Ag(NH_3)_2]MnO_4 \cdot H_2O = 2[AgMnO_2] + 3H_2O + NH_4NO_3 + 2NH_3$$

As can be seen, roughly a quarter of ammonia is oxidized, a quarter turns into ammonium ions, and half is eliminated as ammonia gas. The low decomposition temperature of the $NH_4NO_3$ formed (the second decomposition step) was attributed to the catalytic effect of the silver-manganese oxide on the decomposition of $NH_4NO_3$. Based on the weight loss, the most probable amorphous decomposition product has an "$AgMnO_2$" composition. Although the crystalline $AgMnO_2$ is known as a stable compound of the silver–manganese–oxygen system [78,79], the thermal decomposition of $[Ag(NH_3)_2]MnO_4$ produced a finely dispersed elementary silver and amorphous $MnO_x$ compounds mixture, together with $H_2O$, $N_2$ and NO as gases. The annealing of the solid primary decomposition product at 573 K, the metallic silver reacted with the manganese oxides and resulted in the formation of amorphous silver manganese oxides, which started to crystallize only at 773 K and completely transformed into $AgMnO_2$ at 873 K (Figure 2). This method ensures a simply preparable precursor of ($[Ag(NH_3)_2]MnO_4$) and an easy reaction route to prepare a hardly available phase of pure $AgMnO_2$ [34].

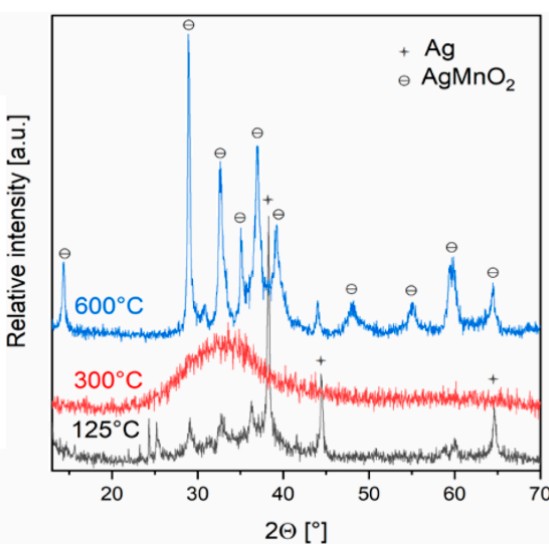

**Figure 2.** Powder XRD of the decomposition product formed from [Ag(NH$_3$)$_2$MnO$_4$]. Reproduced from [34].

The mechanism of the thermal decomposition of [Ag(NH$_3$)$_2$]MnO$_4$ is a complicated process. The overall oxygen amount in the starting complex, taking the AgMnO$_2$ formation into consideration, is only two O atoms/mol. In the first decomposition step, a solid-phase intermediate was the unstable [Ag(NH$_3$)NO$_3$]. This compound decomposes into Ag and MnO$_x$ under further heating, but in the presence of water (aqueous leaching), this intermediate disproportionated into AgNO$_3$ and [Ag(NH$_3$)$_2$]NO$_3$. The hydrolysis of [Ag(NH$_3$)$_2$]NO$_3$ during evaporation of the solvent resulted in insoluble Ag$_2$O and soluble NH$_4$NO$_3$ [71,72], and the AgNO$_3$ and NH$_4$NO$_3$ formed the double salt AgNO$_3$.NH$_4$NO$_3$ isolated in the crystalline form [34,80]. The thermal decomposition process and hydrolysis of the decomposition intermediates are summarized in Scheme 1.

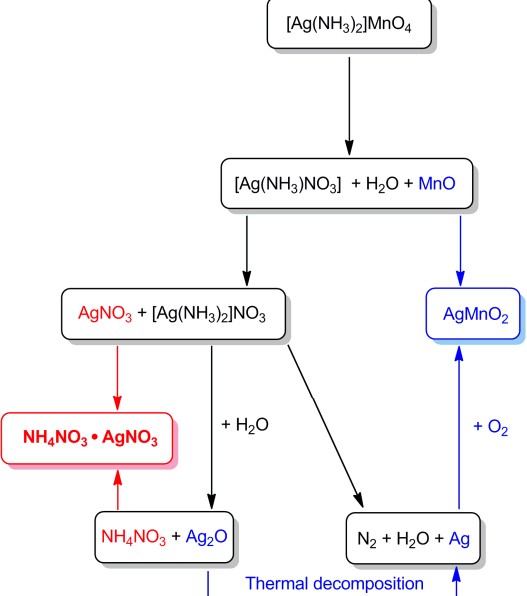

**Scheme 1.** Decomposition scheme of [Ag(NH$_3$)$_2$]MnO$_4$. Reproduced from [80].

### 2.2.2. Diamminesilver(I) Perrhenate

The diamminesilver(I) perrhenate was prepared first by Wilke-Dörfurt in 1933 in the reaction of silver perrhenate with ammonia in an aqueous solution and formed on

cooling as colorless prisms [68]. The crystals are monoclinic and less sensitive to light than the silver perrhenate, and its solubility in an ammonia solution ($d$ = 0.930) at 20 °C was found to be 16.18 g/L. Its density in the solid state is 3.901 g/mL, and its molar volume is 100.5 cm$^3$/mol [68]. No spectroscopic, structural, or thermal decomposition investigations have been conducted on this compound up until now.

2.2.3. [Diamminezinc(II)], [Diamminecadmium(II)], [Diamminecopper(II)], and [Diamminenickel(II)] Perrhenates, [M(NH$_3$)$_2$](ReO$_4$)$_2$ (M = Zn, Cd, Ni)

The [M(NH$_3$)$_2$](ReO$_4$)$_2$ (M = Zn, Cd, Cu, Ni) complexes were prepared as air-stable non-hygroscopic white (Zn, Cd) and green (Cu, Ni) crystalline masses with the isotherm heating of the appropriate [tetraamminemetal(II)] perrhenates at 90 °C (Zn) or 150 °C (Cd, Cu, Ni). These complexes are practically insoluble in water and common organic solvents [81–83]. The powder XRD d-values were determined for the Cu and Ni complexes. However, their crystallographic parameters were not given [81,82].

The IR and Raman spectroscopic studies on the complexes showed the appearance of the symmetric Re-O stretching modes and the splitting of the antisymmetric Re-O stretching modes, which led to the establishment of the coordinated nature of the perrhenate ion in these complexes. The singlet symmetric stretching mode appears as two bands in the Raman spectra of Zn and Cd complexes, and the most probable coordination mode of perrhenate ion in these complexes is unidentate with C$_{3v}$ symmetry [83]. However, the well-separated doublet (20 cm$^{-1}$) nature of the antisymmetric Re-O stretching mode in the Cu complex [81] suggests unidentate or bridging perrhenate anion bonding [81]. The triplet and doublet nature of the antisymmetric Re-O bands of perrhenate ion in the IR and Raman spectra of [Ni(NH$_3$)$_2$](ReO$_4$)$_2$, respectively, were also taken as evidence of the symmetry lowering and coordinated nature of perrhenate ion [82].

The kinetic studies on the non-isothermal decomposition reaction of the Zn complex between 230 and 275 °C resulting in Zn(ReO$_4$)$_2$ showed the reaction order of 0.45 with 97.9 kJ/mol activation energy [83]. The Cu complex decomposition resulted in anhydrous copper perrhenate between 285 and 325 °C, with a reaction order of 0.47 and activation energy of 44.3 kJ/mol [81]. The thermal decomposition of the cadmium complex follows a single mechanism; the activation energy values were found to be 103.1 and 102.7 kJ/mol calculated by the Kissinger–Akahira–Sunose (KAS) and Kissinger methods, respectively [57]. The decomposition of the nickel complex between 529 and 593 K shows a fluctuation of activation energy in the middle of a conversion, which suggests multiple mechanisms. On increasing the conversion temperature, a single mechanism was observed, and the activation energy values were found to be 369.66 and 365.66 kJ/mol, calculated by KAS and Kissinger methods, respectively. The decomposition product was nickel(II) perrhenate [58].

The measured magnetic moment of the Cu complex is 1.81 B.M, and its UV spectrum shows a tetragonal Cu(II) environment (14,000 cm$^{-1}$) [81]. The value of magnetic moment for the Ni complex ($\mu_{eff}$ = 3.07 BM) and its temperature independence shows a similar hexacoordinated nickel(II) environment as was found in the [tetraamminenickel(II)] perrhenate. The d–d transitions in its UV–Vis spectrum are also similar to that of [tetraamminenickel(II)] perrhenate and show a tetragonal (10,400 and 12,800 cm$^{-1}$ ($^3$B$_{1g}$→$^3$B$_{2g}$ or $^3$B$_{1g}$→$^3$E$_g$), 14,300 and 16,700 cm$^{-1}$ ($^3$B$_{1g}$→$^3$A$_{2g}$), 24,400 ($^3$B$_{1g}$→$^3$E$_g$ or $^3$B$_{1g}$→$^3$A$_{2g}$) [82].

*2.3. Triammine Complexes*

Preparation and Properties of [Triamminesilver(I) Permanganate, [Ag(NH$_3$)$_3$]MnO$_4$

The reaction of solid silver permanganate with gaseous ammonia at 10 °C resulted in the triammine complex, [Ag(NH$_3$)$_3$]MnO$_4$, in 72 h [67]. Klobb could not prepare any ammonia complex in the reaction of AgMnO$_4$ and ammonia [69] because this compound does not exist at atmospheric pressure at room temperature [84]. The dissociation curve of [Ag(NH$_3$)$_3$]MnO$_4$ was determined, and the dissociation pressure values at −21, 0, and 10 °C (synthesis temperature) were found to be 97, 330, and 670 mmHg, respectively [85].

### 2.4. Tetraammine Complexes

Among the $[M(NH_3)_4](XO_4)_2$ complexes, only the permanganates and perrhenates of Cu, Zn, Cd, Pd(II), the pertechnetate of Pt(II), and the perrhenates of Pt(II), Ni(II), Co(II) have been prepared until now. There are other known complexes with four ammonia and further other ligands to form an octahedral environment around the central atom, e.g., $[Co(NH_3)_4CO_3]MnO_4$ and $[Ru(NO)(OH)(NH_3)_4](ReO_4)_2$. Single crystal structure determinations were carried out on several representatives, but all known [tetraamminezinc(II)] and [tetraamminecadmium(II)] permanganates and perrhenates, and the [tetraamminecobalt(II)] perrhenate, are isomorphic cubic materials [37,60,86–89] (Table 1). [Tetraamminecopper(II)] perrhenate and the perrhenate and pertechnetate of [tetraamminepalladium(II)] and [tetraammineplatinum(II)] cations, respectively, are isomorphic triclinic (space groups are P-1) crystals [90,91]. The perrhenates of [tetraamminepalladium(II)] and [tetraammineplatinum(II)] cations are also triclinic, but their space groups are P1 [90,92]. The platinum complex has at least two polymorphs, a triclinic and a monoclinic one [93] (Table 1).

**Table 1.** Crystallographic data of $[M(NH_3)_4](XO_4)_2$ [tetraamminemetal(II)] permanganates, pertechnetates, and perrhenates.

| Compound | *T*, K | *a, b, c*, Å | *α,β,γ*, ° | Space Group | Z | *V*, Å³ | $D_{calcd}$, g/mL | Ref. |
|---|---|---|---|---|---|---|---|---|
| M = Cu, X = Mn | 298 | 5.413 9.093 10.749 | 96.18 | P2₁/m | 2 | 526.0 | 2.33 | [94] |
| M = Cu, X = Re | 150 | 6.5167; 6.7790; 7.4627 | 67.336; 80.004 70.687 | P-1 | 1 | | 3.661 | [92] |
| M = Zn, X = Mn | 298 | 10.335 | | F4-3m | 4 | | | [89] |
| M = Zn, X = Re | | 10.53 | | F4-3m | 4 | | 3.60 | [86] |
| | | 10.66 | | F4-3m | 4 | | | [87] |
| M = Cd, X = Mn | 298 | 10.432 | | F4-3m | 4 | | | [60] |
| | | 10.44 | | | | | 2.41 | [86] |
| M = Cd, X = Re | | 10.53 | | F4-3m | 4 | | | [88] |
| | | 10.54 | | | | | | [87] |
| | | 10.67 | | | | | 3.71 | [86] |
| M = Ni, X = Re | | 9.2 5.2 6.7 | | | 1 | | 3.22 | [82] |
| M = Co, X = Re | | 10.54 | | F4-3m | 4 | | 3.56 | [95] |
| M = Pd, X = Mn | | 5.1746 7.5861 7.7217 | 69.313 78.872 76.883 | P1 | 1 | 274.1 | 2.50 | [90] |
| M = Pd, X = Re | | 5.1847 7.7397 7.9540 | 69.531 79.656 77.649 | P-1 | 1 | 290.19 | 4.37 | [90] |
| M = Pt, X = Tc | | 5.179 7.725 7.935 | 69.33 79.74 77.41 | P-1 | 1 | | 3.396 | [91] |

**Table 1.** *Cont.*

| Compound | $T$, K | $a, b, c$, Å | $\alpha, \beta, \gamma$, ° | Space Group | $Z$ | $V$, Å$^3$ | $D_{\text{calcd}}$, g/mL | Ref. |
|---|---|---|---|---|---|---|---|---|
| M = Pt, X = Re | 298 | 5.1847 7.7397 7.9540 | 69.531 79.656 77.649 | P1 | 1 | 290.19 | 4.370 | [92] |
| | | 12.70 8.91 5.09 | 104.1 | C2/m or Cm | 2 | | 4.55 | [93] |

### 2.4.1. Tetraamminecopper(II) Permanganate

[Tetraamminecopper(II)] permanganate was prepared first in the reaction of an ammoniacal copper sulfate solution cooled to 8 °C with potassium permanganate pre-cooled to the same temperature [86,96]. Single crystals could be prepared in this reaction in an ice-cooled bath [94], but the excess ammonia at room temperature resulted in contamination [96]. Starting from [tetraamminecopper(II)] sulfate and potassium permanganate solution, a slower deposition rate of crystals was observed [96]. The best way to prepare pure [tetraamminecopper(II)] permanganate is through the reaction of [tetraamminecopper(II)] sulfate and sodium or potassium permanganate solutions with a 5/2 °C temperature gradient [75]. The reaction product formed by mixing the reactants at room temperature led to a $NH_4MnO_4$-contaminated product due to the temperature-dependent hydrolysis of the complex cation [71,72,75] because of the following: the saturated aqueous $[Cu(NH_3)_4](MnO_4)_2$ solution has a pH value of 9.60, which indicates the partial dissociation of the complex cation into $Cu^{2+}$, $NH_3$ and $[Cu(NH_3)_n]^{2+}$, n = 1–3 species. The dissociation rate increases with increasing temperature. The released ammonia in the solution is partly protonated by the water to form $NH_4^+$ and $OH^-$ ions. Increasing the hydroxide ion concentration causes precipitation of the $Cu^{2+}$ ions formed in complex dissociation equilibrium, removing the $Cu^{2+}$ and $OH^-$ ions from the solution due to the low solubility product of $Cu(OH)_2$ ($K_{sp} = 4.8 \times 10^{-20}$). It shifts the complex decomposition and ammonia protonation equilibrium with the accumulation of free ammonium and permanganate ions, which, due to the low solubility of $NH_4MnO_4$ [65], results in co-deposition of $NH_4MnO_4$ together with $[Cu(NH_3)_4](MnO_4)_2$ (the solubility product of $[Cu(NH_3)_4](MnO_4)_2$ is $L = 7.81 \times 10^{-3}$ [71]):

$$[Cu(NH_3)_4](MnO_4)_2 + 2H_2O = Cu(OH)_2 + 2NH_3 + 2NH_4MnO_4$$

This equilibrium can be completely shifted to the right with the removal of the ammonia from the equilibrium. The ammonia vapor pressure is higher than that of the water. Furthermore, it depends on the temperature, and thus the outer pressure decreasing (vacuum) or increasing temperature (heating) can complete the hydrolysis process. The remaining ammonium and permanganate ions were crystallized out as ammonium permanganate [73]. This reaction route was used to prepare single crystals of highly pure alkali metal-free ammonium permanganate [72,75].

[Tetraamminecopper(II)] permanganate is a purple-black crystalline powder [96], or violet crystalline material [94], slightly soluble in water and in its aqueous solutions gradually decomposes, and the initially beautiful purple color of the solution disappears, and manganese oxide is deposited [96]. It is soluble without similar decomposition in diluted sulfuric acid [96]. It also dissolves in polar organic solvents such as DMF and $Ac_2O$ but is insoluble in non-polar solvents like hydrocarbons and chlorinated solvents [75]. In its DMF solution, [tetraamminecopper(II)] permanganate dissociates completely into cation and anion (the antisymmetric stretching Mn-O mode of permanganate ion appeared as a sharp intensive singlet band at 900 cm$^{-1}$), whereas the Cu-N region of the IR spectrum showed a very wide band system due to the solvation/ligand exchange of the complex cation with the DMF solvent [75]. The DMF solution of $[Cu(NH_3)_4](MnO_4)_2$ has an intense purple color which disappears within half an hour [75], and $MnO_2$ formation was observed.

In DMSO, it dissolves with the formation of a green compound that decomposes fast on standing [75].

It is stable in the dry state for several weeks, but while remaining in a wet or impure state for longer periods, it easily decomposes with $MnO_2$ formation, especially above 10 °C, and the sunlight increases the decomposition rate [94,96]. It detonates under the shock of the hammer, heating or crushing in a mortar [86,96], and fusing with releasing ammonia, producing a cloud of very finely divided oxides and voluminous lightweight and contoured ash [96].

Its prismatic twinned single crystals were grown by slow evaporation of a saturated aqueous solution over concentrated $H_2SO_4$ at ~278 K, $d_{exp}$(flotation) = 2.39 g/mL [94]. Its monoclinic elementary cell (Table 1) contains two isolated $[Cu(NH_3)_4(MnO_4)_2]$ units with distorted octahedral copper(II) (4 + 2) coordination with four ammonia nitrogen atoms in the equatorial and two permanganate oxygen atoms in the axial sites [94].

Due to the two crystallographically nonequivalent permanganate ions, the IR and Raman bands are doubled, although among the 2 × 3 antisymmetric Raman bands, only a quintet (instead of two separated triplets) can be seen due to the overlapping (Figure 3). Since the F2 mode (antisymmetric Mn-O stretching) of permanganate ion is a triply degenerate mode, the presence of more than three bands unambiguously shows the crystallographic inequivalence of the two permanganate ions in $[Cu(NH_3)_4](MnO_4)_2$ even without single-crystal studies.

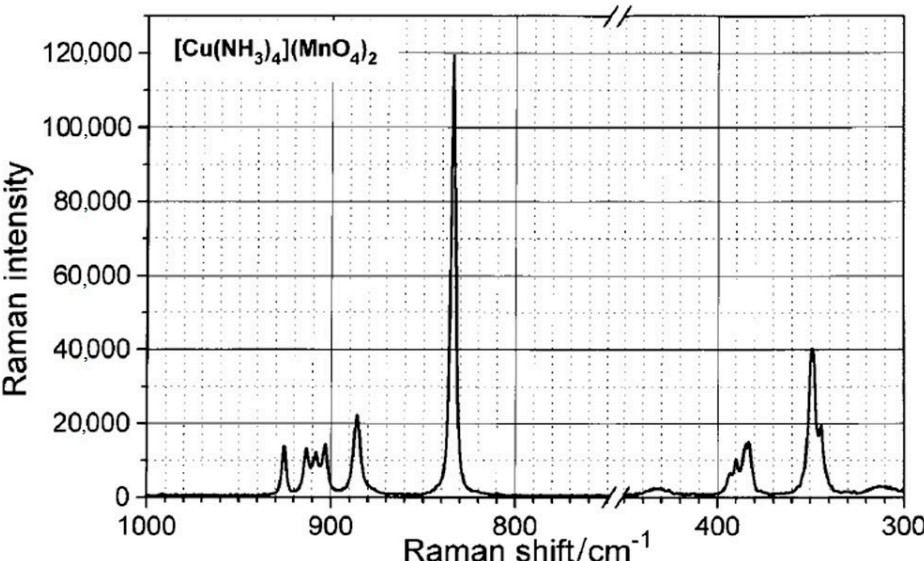

**Figure 3.** Raman spectrum of $[Cu(NH_3)_4](MnO_4)_2$. Reprinted/adapted with permission from [75].

All of the IR and Raman bands belonging to the cation and anion modes were completely assigned. The shift and splitting of the hydrogen-bond sensitive rocking $NH_3$ mode ($\rho(NH_3)$) in the IR spectrum of [tetraamminecopper(II)] permanganate indicate the presence of weak N-H ... O-Mn hydrogen bonds [75]. The ESR $g$-factors ($g_{zz}$ = 2.273, $g_{xx} = g_{yy}$ = 2.090) are typical according to the O-ligation with a square-planar Cu environment. The sharpness of the parallel and perpendicular ESR bands of $[Cu(NH_3)_4](MnO_4)_2$ and the lack of Cu hyperfine structure show that the exchange interactions between the magnetically equivalent copper centers are stronger than the dipole couplings [75].

[Tetraamminecopper(II)] permanganate has unique thermal properties. Its heat of formation is 355 kcal/mol and burns easily under oxygen pressure according to the following equation [97]:

$$[Cu(NH_3)_4](MnO_4)_2 = Cu(liq.) + 0.66Mn_3O_4 + 5.34H_2O + 0.44NH_3 + 1.78N_2$$

The heat of combustion is 152 kcal/mol, and the calculated combustion temperature is 1500 K. It starts to burn at 8 technical atm pressure with a dark red glow. It is a fast-burning material, $u_m$ = 16 g/cm$^2$.s.gauge atm, $t_{del}$ = <1 s at 280 °C [97]. The preliminary TG studies in air showed that [tetraamminecopper(II)] permanganate explodes at ~363 K, giving a mixture of Cu and Mn oxides. It decomposes under N$_2$ in at least two steps. Based on the weight losses, the decomposition mechanism was summarized as follows [94]:

$$[Cu(NH_3)_4](MnO_4)_2 \rightarrow Cu(MnO_4)_2 \rightarrow CuMn_2O_4 -$$
$$-4NH_3, 3013–423 \ K \ –2O_2, 423–773 \ K$$

The intermediate 'Cu(MnO$_4$)$_2$' was amorphous, whereas the CuMn$_2$O$_4$ was found to be crystalline [94]. Further studies that used combined methods (TG-MS, DSC) pointed out a much more complicated decomposition mechanism [75].

DSC studies showed a strongly exothermic decomposition reaction (the ammonia ligand loss is expected to be endothermic) in both steps, and the IR results of the decomposition intermediates did not show the presence of permanganate ion, whereas the IR spectrum of the intermediate formed at 250 °C contained a sharp, unexpected peak at ~2200 cm$^{-1}$. The shift of the Mn-O antisymmetric stretching modes to the low wavenumber region showed a strong decrease in the oxidation number of manganese [75]. These results completely coincide with the formation of [Cu(NH$_3$)$_2$](MnO$_4$)$_2$ and Cu(MnO$_4$)$_2$ intermediates given previously.

TG-MS and TG-gas titrimetric studies showed that in the two well-defined decomposition steps of [tetraamminecopper(II)] permanganate, 2 mol of ammonia, water, and N$_2$O are formed without oxygen evolution. The two ammonia molecules were formed in the first step, together with H$_2$O evolution, and in the second step, N$_2$O and H$_2$O could be detected as main products, but without any ammonia evolution. The decomposition residue is amorphous; its formula corresponds to the CuMn$_2$O$_{4+x}$ stoichiometry. It does not dissolve in nitric acid; thus, it is not the stoichiometric mixture of CuO and Mn-oxides. Heating of the amorphous decomposition residue until 500 °C resulted in cubic CuMn$_2$O$_4$ spinel. The IR spectra of the decomposition intermediate showed the presence of ammonium nitrate, confirmed by the XRD and IR of the evaporation residue crystallized out from the aqueous leachate. The formation of N$_2$O and H$_2$O in the second step is attributed to the decomposition of NH$_4$NO$_3$, and the sharp peak in the IR of the decomposition intermediate belongs to the gas inclusion of N$_2$O [75]. The two-step process was described as follows [75]:

$$[Cu(NH_3)_4](MnO_4)_2 = CuMn_2O_4 + 2NH_3 + NH_4NO_3 + H_2O$$

$$NH_4NO_3 = N_2O + 2H_2O$$

In the first decomposition step, the oxidation of one ammonia ligand into nitrate and H$_2$O is a strongly exothermic process, and its reaction heat overflows the endothermicity of the two residual ammonia ligand losses. The reaction starts at 65 °C, which is lower than the temperature of the ammonia ligand loss of [tetraamminecopper(II)] cation; thus, the ammonia-permanganate redox reaction takes place in solid state [75].

The temperature-controlled decomposition of [tetraamminecopper(II)] permanganate in CHCl$_3$ and CCl$_4$ at 61 and 77 °C, respectively, resulted in the formation of amorphous copper manganese oxide and ammonium nitrate mixtures. The oxygen surplus in the CuMn$_2$O$_{4+x}$ oxides varied between $x$ = 0 and 0.35. The formation of (Cu,Mn)$_2$O$_3$ and (Cu,Mn)$^{T-4}$(Cu,Mn)$^{OC-6}_2$O$_4$ oxides, their crystallite size, and catalytic activity strongly depend on the heating temperature (CHCl$_3$ or CCl$_4$) or the removal of ammonium nitrate (aq. washing or heat treatment (Scheme 2) [28]).

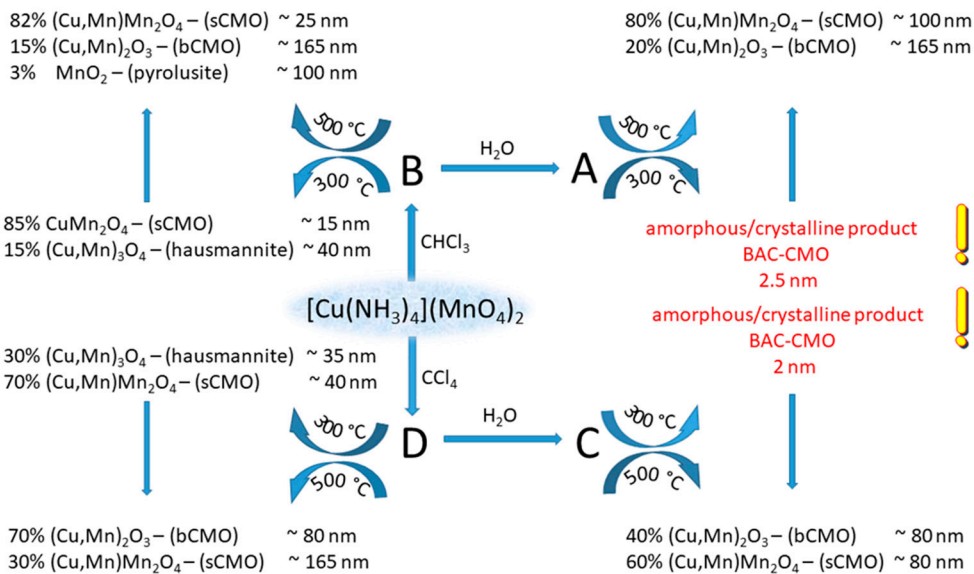

**Scheme 2.** Temperature-controlled decomposition scheme of [tetraamminecopper(II) permanganate]. Reproduced from [28].

These copper manganese oxides were proved to be catalytically active in CO oxidation; thus, [tetraamminecopper(II) permanganate] is a potential candidate in the preparation of Hopcalite-like catalysts [28].

The oxidative deoximation of aldoximes and ketoximes and oxidative regeneration of phenylhydrazones by [tetraamminecopper(II)] permanganate in 1:1 aqueous acetic acid resulted in the appropriate oxo-compounds [94,98]. Benzaldehyde and acetophenone oximes are deoximated at room temperature with 82 and 81% yield, respectively. The oxidative regeneration of acetaldol phenylhydrazone resulted in a three-electron reduction of [tetraamminecopper(II)] permanganate with the formation of $Mn^{IV}$. No influence of acrylonitrile on the reaction and no acrylonitrile polymerization (lack of free radicals) were observed [55,56].

The kinetics of the oxidative regeneration of oximes (278–298 K) and phenylhydrazones (288–318 K) of H-C(=O)-$R^1$ aldoximes ($R^1$ = H, Me, Et, Pr, i-Pr, ClCH$_2$, Ph) and $R^1$-C(=O)-$R^2$ ketoximes ($R^1$ = Me, $R^2$ = Me, Et, Ph and $R^1$ = $R^2$ = Et) were studied in 1:1 aq. acetic acid. The reactions are first order for the organic components and oximes and [tetraamminecopper(II)] permanganate as well. The oxidation rate of keto-derivatives is slower than that of aldehyde-derived compounds. The substituent-dependent regeneration reaction rates can be described by the Pavelich–Taft dual substituent-parameter equation. The low positive values found for the polar reaction constants indicate a nucleophilic attack by a permanganate-oxygen on the double-bond carbon atom. The low activation enthalpy values indicate that the bond cleavages and bond formations are almost synchronous. The large negative values of activation entropies support the formation of a rigid cyclic activated complex. The steric hindrance of the alkyl group has an influence on the reaction. The reaction rate-determining step is the formation of an acyclic intermediate [55,56].

The [tetraamminecopper(II)] permanganate oxidizes benzyl alcohol in CHCl$_3$ solution into benzaldehyde with 67% yield in 3 h reflux, but only 4% benzonitrile was formed. Increasing the reflux temperature (CCl$_4$) increased the nitrile yield to 14% in 3 h. This indicates that the ammonia could not be liberated easily from the coordination sphere to react with the aldehyde formed. The addition of coordinating solvents to substitute ammonia in the coordination sphere of the complex cation (DMF or CH$_3$CN) suppressed the nitrile formation due to the stable solvated ammine complex formation [65]. It shows that the stability of the NH$_3$ source is a key factor in the ammoxidation reactions of benzyl alcohol into benzonitrile because the ammonium permanganate [99] or [hexakis(urea)iron(III)]

permanganate (urea acts as ammonia precursor) [11] results in much more benzonitrile formation than the [tetraamminecopper(II)] permanganate [65].

The oxidation of benzyl alcohol and its ortho (Me, OMe, NO$_2$, COOMe, F, Cl, Br, I, CN, NHAc, SMe, CF$_3$), meta (Me, OMe, F, Cl, NO$_2$, CF$_3$, COOMe, Br, NHAc, CN, SMe) and para (Me, OMe, Cl, Br, F, NO$_2$, COOMe, CF$_3$, CN, SMe, NHAc, NMe$_2$) monosubstituted benzyl alcohols by [tetraamminecopper(II)] permanganate in aqueous acetic acid resulted in the formation of the corresponding benzaldehydes. The kinetics of these reactions were measured between 288 and 318 K; moreover, the rate constants and activation parameters were calculated. The reactions are first order regarding the oxidant, substrates, and hydrogen ions. The oxidation of PhCD$_2$OH exhibits a substantial temperature-dependent kinetic isotope effect ($k_H/k_D$ at 298 K was found to be 5.83). The reaction rate increases with an increase in the polarity of the solvent. The oxidation rates of the meta- or para- and the ortho-substituted benzyl alcohols correlated in terms of Charton's triparametric LDR and tetraparametric LDRS equations, respectively. The oxidation of para-substituted benzyl alcohols is more sensitive to the delocalization effect than that of the ortho- and meta-substituted derivatives, which show a great dependence on the field effect. The positive ρ values suggest the presence of electron-deficient reaction centers in the rate-determining step. The ortho substituents show steric acceleration [76].

### 2.4.2. Tetraamminezinc(II) and [Tetraamminecadmium(II)] Permanganates

[Tetraamminezinc(II)] and [tetraamminecadmium(II)] permanganate were prepared first by Klobb as a fine purple powder in the reaction of 0.2 M zinc and a cadmium sulfate, ammonium hydroxide, and saturated potassium permanganate solution at 10 °C. To avoid contamination, the Zn complex that formed must be filtered very quickly [86,96]. The Cd complex cannot be dried over lime or sulfuric acid without decomposition, but drying over P$_2$O$_5$ resulted in black crystals/purple [tetraamminecadmium(II)] permanganate in 48 h [96]. The microcrystalline [M(NH$_3$)$_4$](MnO$_4$)$_2$ (M = Zn, Cd) complexes were prepared in a pure state in the reaction of the saturated aqueous [tetraamminezinc(II)] or [tetraamminecadmium(II)] sulfate and KMnO$_4$ solutions with +5/+2 °C temperature gradient [37,60,86]. [Tetraamminezinc(II)] and [tetraamminecadmium(II)] permanganates [86,96], $d_{exp}$ = 2.27 g/mL (M = Zn) [86]. They explode upon rubbing or crushing in a mortar [86,96], under the shock of the hammer, or upon heating, releasing ammonia and producing a cloud of very finely divided oxides. They give a brown insoluble powder after 1–2 h (M = Zn) or a few days (M = Cd) storage at room temperature [96]. In a wet form, the Zn complex easily decomposes if exposed to light [96] but is stable in dry conditions at room temperature [37].

Both complexes are slightly soluble in water (0.91 g/100 mL H$_2$O at 19 °C for the Zn compound [37]), but their aqueous solutions lose their beautiful purple color, and manganese oxide is quickly deposited [60,61]. Their hydrolysis reaction proceeds in aq. solutions, with the formation of ammonium permanganate and zinc(II) or cadmium(II) hydroxide [60]. They dissolve in diluted sulfuric acid, and no sign of similar decomposition could be seen [96].

The powder X-ray data of [tetraamminezinc(II)] and [tetraamminecadmium(II)] permanganates (*d* values and their relative intensities), including Müller indices, have been reported [86]. Single crystal studies showed some additional weak reflections indicating a face-centered cubic supercell with Z = 32 and *a* = 20.62 Å (M = Zn) and *a* = 20.88 Å (M = Cd) values. These large cells show a slight distortion of the tetrahedral units since the intensities of the weak additional reflections were calculated to be zero, assuming an exactly tetrahedral environment. The space group for the structures with a larger cell could not be determined exactly but it is probably T$_d$$^2$-F4-3m or F4-3c-T$_d$ [86]. Due to the lack of good-quality single crystals, their structures were solved from PXRD data with Rietveld refinement using the atomic coordinates of the isostructural [Zn(NH$_3$)$_4$](ClO$_4$)$_2$ [37,60]. [Tetraamminezinc(II)] and [tetraamminecadmium(II)] permanganates crystallize in a closely packed cubic structure (Table 1) and build up a three-dimensional M-N-H....O-Mn hydrogen-bonded network with block-like structural motifs of four M(NH$_3$)$_4$$^{2+}$ and four MnO$_4$$^-$

ions (M = Zn, Cd). Only one of the two permanganates of each complex takes part in the building up of the 3D network; the second kind of permanganate ion is captured in the cavities enclosed by the tetramer building blocks of the network [37,60]. There are significant differences between the strength of the N-H...O-Mn hydrogen bonds of the 3D network forming and the cavity-embedded permanganate ions. The hydrogen bonds between the ammonia hydrogens and the cavity-embedded permanganate ions hinder the free rotation of the embedded permanganate ion, although its rotational freedom is higher than that of the network-fixed permanganate ion [37,60].

The IR and Raman bands of [tetraamminezinc(II)] and [tetraamminecadmium(II)] permanganates were completely assigned, and the cation and anion modes were listed. The splitting of antisymmetric Mn-O stretching bands (F2) could not be explained well by simple site-symmetry considerations based on the $T_d{}^2$ space group; therefore, dynamic effects through the interaction of neighboring ions in a Z = 4 unit cell (factor group splitting neglected in site symmetry considerations) are noticeable [86].

Due to the presence of two crystallographically different tetrahedral permanganate ions in the lattice, the IR and the Raman bands belonging to the Mn-O modes are doubled. The $MN_4$ skeleton (M = Zn, Cd) of the complex cation also has tetrahedral geometry. The factor group analysis indicated that the degeneration of the antisymmetric modes (F2) in this crystallographic environment due to the symmetry relations should not be ceased, and the symmetric IR modes should not appear. The appearance of the IR forbidden symmetric ($\nu_s$ and $\delta_E$) modes and the splitting of the triply degenerated (F2) antisymmetric modes ($\nu_{as}$ and $\delta_{as}$) was attributed to the orientation effects of a cavity-embedded permanganate ion, which is strengthened with decreasing temperature [37,60]. The increase of the $\nu_{as}$(Mn-O)/$\nu_s$(Mn-O) integrated intensity ratio values by decreasing the temperature from 293 to 173 K indicates that the permanganate-ion orientation is frozen, whereas the dynamic lattice distortion, due to slowing down the anisotropic thermal motions, changes with an opposite sign under cooling. The IR bands belonging to cavity-embedded permanganate were identified [37,60].

The thermal decomposition of [tetraamminezinc(II)] and [tetraamminecadmium(II)] permanganates under an inert atmosphere proceeds in 2 + 1 steps with the formation of an amorphous decomposition product with an $MMn_2O_{4+x}$ ($x$ = 0–0.35) formula (M = Zn, Cd), which transforms at 500 °C. The peak temperatures of the first decomposition step strongly depend on the heating rate and vary between 100 and 130 °C. Two molecules of ammonia are released only in the first decomposition steps at 100 °C, the other two transform into ammonium nitrate and water. No $O_2$ evolution occurs. In the second decomposition step, the ammonium nitrate decomposition was observed, which is catalyzed by the presence of $MMn_2O_4$ (M = Zn, Cd) spinels at 230 and 224 °C, respectively.

The main decomposition processes are the following:

$$[M(NH_3)_4](MnO_4)_2 = MMn_2O_4 + NH_4NO_3 + 2NH_3 + H_2O \ (M = Zn, Cd)$$

$$NH_4NO_3 = N_2O + 2H_2O$$

All decomposition steps are exothermic; the overall reaction heats are ∆H = −169 and −318 kJ/mol, which shows that the reaction heats of the ammonia-permanganate reactions in these complexes are higher than the energy demands of the partial de-ammoniation reactions [37,60].

In toluene as heat-convection media, the reaction temperature is controlled at 110 °C by the evaporation of toluene (reflux). Leaching of the amorphous decomposition residue with water (removal of $NH_4NO_3$) resulted in amorphous $MMn_2O_{4+x}$ ($x$ = 0–0.35) (M = Zn, Cd) compounds, and the leaching solutions evaporation produced crystalline ammonium nitrate [37,60].

2.4.3. [Tetraamminecopper(II)], [Tetraamminezinc(II)], and [Tetraamminecadmium(II)] Perrhenates

[Tetraamminecopper(II)] perrhenate was prepared first by Briscoe et al. [100] by adding aq. concentrated ammonium hydroxide or gaseous ammonia to a concentrated hot copper(II) perrhenate solution until the formed precipitate was completely re-dissolved, and then the deep blue crystals were crystallized out after cooling the solution [100]. The copper(II) perrhenate prepared in situ from $CuCO_3$ and $HReO_4$ was mixed directly with concentrated ammonia [81]. [Tetraamminecopper(II)] perrhenate was also prepared in the metathesis reaction of ammoniacal copper sulfate with ammonium perrhenate or perrhenic acid between 20 and 60 °C [101]. Neither the ammonia nor the perrhenic acid excess caused the formation of any other insoluble phase. The optimal concentration of each reactant to prepare the maximal yield of $[Cu(NH_3)_4](ReO_4)_2$ has been determined [101]. It was also prepared from a hot aq. solution of copper(II) acetate mixed with 1:1 aq. ammonia to obtain pH 11–12, with two equivalents of concentrated aq. solution of sodium perrhenate. On cooling, a dark lilac crystalline precipitate formed with a yield of ~77% [89].

[Tetraamminezinc(II)] and [tetraamminecadmium(II)] perrhenates were synthesized in an analogous way as the above-mentioned copper(II) complex from the appropriate metal perrhenates and aqueous ammonia [57,68,88]. Zagorodnyaya et al. prepared [tetraamminezinc(II)] perrhenate with the addition of zinc sulfate, then 0.025–0.1 M ammonium perrhenate solutions to a 20% excess of aq. ammonia at room temperature [102]. [Tetraamminecadmium(II)] perrhenate and its deuterium and $^{116/110}$Cd isotope-containing samples were prepared in a similar way from $Cd(ReO_4)_2$ and $^{110/116}Cd(ReO_4)_2$ (prepared by the reaction of $CdCl_2$, $^{110/116}CdCl_2$ and $AgReO_4$, respectively) with aq. ammonia or deuterated ammonia in heavy water [103]. It also formed during the processing of rhenium-containing lead dust, when rhenium generally follows the cadmium during the extraction of Zn and Cd chloride and sulfate-containing solutions with trialkyl amines with subsequent re-extraction with aq. ammonia [104]. The bluish violet (Cu) and colorless (Zn, Cd) microcrystalline substances were purified by recrystallization from warm concentrated ammonia [68,81].

[Tetraamminecopper(II)] perrhenate is a blue [86] or bluish violet [94,103], and the Zn and Cd complexes are white powders or colorless crystals [68,83,86]. They are non-hygroscopic, insoluble in water, and common organic solvents [68,83,102]. The Cu complex is stable in air until 100 °C [100], starts to decompose at 140 °C, and on further heating in air, turns into pale green materials and then melts with darkening and completely decomposes [81,100]. It forms regular prismatic anisotropic violet-blue single crystals with medium to high birefringence. Distinct pleochroism is from pale blue to dark purple, and the optical phase is biaxial and negative. Refractive indices: $n_p$ = 1.635–1.64; mboxemph$n_g$ = 1.655–1.660 [101].

Both the Zn and Cd complexes consist of regular cubes, but the Zn complex forms octagons or rectangles as well [83,102]. The single crystalline Zn complex is isotropic (refractive index is n = 1.627) [102], and its density and molar volume are $d$ = 3.608 g/mL and 175.75, respectively [68,86]. The pycnometric density and molar volume of the Cd complex are $d$ = 3.714 (25 °C) g/mL [69] or 3.72 g/mL [86] and 183.5 [68]. There is controversial information about the solubility of [tetraamminezinc(II)] and cadmium perrhenates, which are given as insoluble material in water [83], but Zagorodnyaya produced the solubility of Zn complex as 6.23 g/100 g water [102]. The solubilities of the Zn and Cd complexes in ammonia ($d$ = 0.930) at 20 °C is 1.852 g/L [68] (increases from 0.128 to 0.359 g/100 g solution, increasing the ammonia concentration from 1.2 to 12.0 M [102]) and at 11 °C is 0.37 g/L, respectively [68].

The powder-X-ray diffractogram of the Cu, Cd, and Zn complexes, including the $d$-values and intensity data, were discussed [81,86,95,101,102], and the Miller indices were given for the Zn and Cd complexes [86,95]. Their single crystal studies showed some weak additional reflections as those were found in the case of isomorphous permanganate salts (face-centered cubic supercell with $a$ = 21.06 Å, Z = 32, $T_d^2$-F4-3m) [86]. The lattice constant of the Cd complex agrees satisfactorily with the calculated one from the experimental

density ($d^{25}_4$ = 3.714, $a_0$ = 10.66 Å) when four molecules are taken to the unit [88]. The cadmium complex has a first-order order–disorder type phase transition at 368 K with 2.0 kJ/mol enthalpy change and 8 K hysteresis during the DSC measurements [57].

The IR bands of the [tetraamminecopper(II)] perrhenate and their assignments were discussed in detail. The forbidden symmetric stretching Re-O mode of perrhenate ion appears in the IR spectrum together with a well-separated doublet (10 cm$^{-1}$) of the antisymmetric Re-O stretching mode suggesting unidentate or bridging perrhenate anion bonding [81,101]. The magnetic moment value is $\mu_{eff}$ = 1.84 B.M, and the UV spectrum shows bands characteristic of a tetragonal Cu environment (16,000 and 14,000 cm$^{-1}$) [81].

The IR bands of [tetraamminezinc(II)] and [tetraamminecadmium(II)] permanganates were assigned completely [86,104], including measurements on $^{14}$N/$^{15}$N and $^{110/116}$Cd isotope substituted derivatives to assign the ZnN$_4$ and CdN$_4$ skeletons, and on H/D isotope substituted derivatives to the unambiguous assignation of the ammonia ligand modes [103,105]. The wavenumber shifts due to $^{14}$N/$^{15}$N isotope substitution were varied between 2.5–9.0 cm$^{-1}$, and the highest shifts were found for antisymmetric Zn-N stretching (9 cm$^{-1}$) and for the symmetric HNH deformation modes (7 cm$^{-1}$) [103]. The $^{110/116}$Cd isotope substitution does not cause isotope shift in the Cd-N symmetric stretching and bending modes in the Raman spectra, whereas the H/D substitution caused 27 and 16.5 cm$^{-1}$ shift in the IR, and 6 cm$^{-1}$ shift in the Raman spectra. The antisymmetric stretching and bending modes showed 2.0 and 0.5 cm$^{-1}$ ($^{110/116}$Cd) and 23 or 12.5 cm$^{-1}$ shifts (H/D) isotope shifts in the IR spectra, respectively [105].

The symmetric Re-O stretching mode of perrhenate-ion is absent, and the antisymmetric Re-O stretching mode is not split in the IR spectra of the Zn and Cd complexes, which indicates the non-coordinating nature of perrhenate ion in this complex [83]. However, Müller et al. [86] found splitting of antisymmetric Re-O stretching bands (F2) in both IR spectra, which could not be explained by simple site-symmetry considerations based on T$_d^2$ space-group; therefore, dynamic effects through the interaction of neighboring ions in a $Z$ = 4-unit cell (factor group splitting neglected in site symmetry considerations) are noticeable [86]. Experimental IR and Raman spectra were measured and compared with the results of quantum-chemical calculations [57]. The complete assignments of the IR and Raman bands have been given [57]. The low-temperature IR and Raman scattering measurements revealed that the coordinated ammonia ligands perform fast reorientation motions below 368 K. This motion is slowed down at around 40 K. The estimated activation energy for this motion was found to be ~4 kJ mol$^{-1}$ from both the IR and Raman measurements. It was confirmed by quasi-elastic neutron scattering measurements, which confirmed that the ammonia ligands reorientate even in low temperatures as well. These motions are probably jumped around a 3-fold symmetry axis [57].

The crystal structure of [tetraamminecopper(II)] perrhenate was determined at 150 K [89], and it is isostructural with [M$^A$(NH$_3$)$_4$](M$^B$O$_4$)$_2$ (M$^A$ = Pt, Pd; M$^B$ = Re, Mn) (Table 1). The Cu atom (in the symmetry center) is coordinated with four N atoms located at the vertices of a square. The perrhenate anion is slightly distorted. There are N–H . . . O-Re hydrogen bonds, the shortest one being 2.179 Å. Translation sub-lattice isolation technique resulted in sub-cell parameters $a_c$ = 6.52 Å, $b_c$ = 5.14 Å, $c_c$ = 3.90 Å, showing that it can be conventionally considered hexagonal, and it was identified hexagonal layers formed by metal atoms, which are repeatedly with $c_c$ = 3.90 Å [89].

[Tetraamminecopper(II)] perrhenate hydrolyzes in its aqueous solutions with the formation of a blue precipitate (Cu). The blue crystalline phase that formed from the Cu complex is not the expected Cu(OH)$_2$ as that which was found in the case of [Cu(NH$_3$)$_4$](MnO$_4$)$_2$ [75]. This phase is orthorhombic, and its lattice parameters are $a$ = 11.276 Å, $b$ = 15.460 Å, $c$ = 16.865 Å. The intensities of IR bands are weak compared to the OH bands found at 704 cm$^{-1}$ (it is located at 696 cm$^{-1}$ in Cu(OH)$_2$). The heating of the solution containing this phase causes CuO to appear at 50 °C. The compound contains copper (59.5%) and a small amount of rhenium (1.75% and ammonia (0.7%). The bands at 1500, 1464, and 1372 cm$^{-1}$ might belong to N-O or N-H bond-containing species as well [101]. The

hydrolysis process of the complexes is completed at 100 °C as in the case of permanganate and perchlorate complexes (M = Cu, Zn, Cd) with the formation of $M(OH)_2$ (M = Cu, Zn, Cd) precipitates and simple ammonia liberation [71,72,75,102,104,106]. Increasing the temperature, [tetraamminecopper(II)] perrhenate also follows this kind of hydrolysis process because above 50 °C, CuO (as a dehydration product of $Cu(OH)_2$) appears as the main reaction product.

$$[M(NH_3)_4](ReO_4)_2 + 4H_2O = M(OH)_2 + 2NH_4OH + 2NH_4ReO_4$$

$$M(OH)_2 = MO + H_2O, \ M = Cu, Zn, Cd$$

The presence of ammonia prevents the hydrolysis equilibrium, and [tetraamminecopper(II)] perrhenate dissolves in 1.2–12 M ammonia solutions without decomposition. Its solubility is between 0.898–3.11 g/100 g solution.

The hydrolysis of [tetraamminezinc(II)] and [tetraamminecadmium(II)] perrhenates can be prevented even in 1.2 M ammonia solutions, and the original salts can be crystallized out [102,106]. Removing ammonia from the hydrolysis equilibrium [72] with mineral acids (HCl, $H_2SO_4$, or $HNO_3$) or acetic acid shifts the hydrolysis equilibrium into the direction of ammonium perrhenate formation. The acid concentrations above 0.1 M cause complete decomposition [101,102,106]:

$$[M(NH_3)_4](ReO_4)_2 + 2HX = MX_2 + 2NH_4ReO_4 + 2NH_4X$$

$$X = Cl, \ OAc, \ NO_3 \ or \ \frac{1}{2}SO_4 \ M = Cu, Zn, Cd$$

The effects of temperature and acid concentrations were optimized, including adding an excess of ammonium sulfate as a salting-out agent both for the aqueous (100 °C) hydrolysis and sulfuric acid neutralization processes [101,102,106].

The thermal decomposition of the $[M(NH_3)_4](ReO_4)_2$ (M = Cu, Zn, Cd) complexes in an inert atmosphere show a decomposition step at 175–225, 150–195, and 100–150 °C with the formation of $[M(NH_3)_2](ReO_4)_2$ compounds with 0.37, 1.27, and 0.42 reaction order 67.2 kJ/mol, 47.3 kJ/mol, and 28.9 kJ/mol activation energy, respectively [81,83]. Hetmanczyk et al. [57] determined the kinetic parameters for the first decomposition step of the cadmium complex resulting in the formation of diamminecadmium(II) perrhenate. The decomposition step follows a single mechanism, and the activation energy values were found to be 97.7 and 101.7 kJ/mol, as calculated by Kissinger–Akahira–Sunose and Kissinger methods, respectively [57].

The thermal decomposition of the complexes in an air atmosphere, however, looks more complicated. The decomposition between 130–245, 129–360, and 250–360 °C takes place with the formation of ammonium perrhenate and hydrated copper, zinc, and cadmium perrhenates, respectively. The next decomposition steps belong to the decomposition of ammonium perrhenate and metal perrhenates, with elimination and redox reactions of rhenium heptoxide with $ReO_2$ and $ReO_3$ formation and with their re-oxidation into $Re_2O_7$ [107–109]. The final decomposition residues are CuO, ZnO, and CdO [101,102,104,107,110].

2.4.4. [Tetraamminenickel(II)] and [Tetraamminecobalt(II)] Perrhenates, $[M(NH_3)_4](ReO_4)_2$ (M = Ni, Co)

[Tetraamminenickel(II)] perrhenate was isolated as a decomposition intermediate of [hexaamminenickel(II)] perrhenate on standing in air [58,111], whereas the cobalt complex was precipitated, as a bright violet cube, from a hot aq. cobalt(II) perrhenate solution by passing ammonia gas [68,100]. Excess of ammonia resulted in the formation of a brown precipitate [100]. Under an inert atmosphere, the concentrated solution of cobalt perrhenate containing a small amount of hydroxylamine hydrochloride and ammonia gas resulted in a bright red precipitate, which turned into a magnificent, crimson-colored regular tetrahedron



containing crystalline mass upon shaking ($d$ = 3.428 g/mL; mol-volume is 183 $cm^3$/mol) without formation of the brown oxidation by-product [68].

The [tetraamminenickel(II)] perrhenate is stable at 100 C, but with strong heating in air, decomposes with nickel oxide formation [100]. It is non-hygroscopic and insoluble in water or organic solvents. The violet crystals of $[Co(NH_3)_4](ReO_4)_2$ are stable in air and can be washed with ammonia-containing water without decomposition. However, its treatment with ammonia-free water resulted in an insoluble bright green material (probably a basic perrhenate) [100].

The powder XRD of $[Ni(NH_3)_4](ReO_4)_2$ shows an orthorhombic lattice (Table 1); the experimental density was found to be 2.96 g/mL [82]. $[Co(NH_3)_4(ReO_4)_2]$ is cubic (Table 1), $d_{exp}$ = 3.43 g/mL [95]. It is isostructural with the analog $[M(NH_3)_4](XO_4)_2$ and $[M(NH_3)_4](OSO_3N)_2$ (M = Cd, Zn, X = Re, Mn) compounds. Single-crystal diffraction measurements showed additional weak reflexes, which were not measurable in the powder records. It shows the presence of a possible cubic flat-centered superstructure with $a$ =21.08 Å, Z = 32, and $T_d^2$-F4-3c [95].

The triplet and doublet nature of the antisymmetric Re-O band of perrhenate ion in $[Ni(NH_3)_4](ReO_4)_2$ and $[Co(NH_3)_4](ReO_4)_2$, respectively, were taken as evidence of the perrhenate ion coordination [82]. Since the $\nu_3$(Co-N) mode did not split in the IR spectrum of the Co complex (the $\nu_3$(Re-O) is a doublet), the distortion of the $ReO_4$ tetrahedron was declared to be higher than that of the $CoN_4$ tetrahedron in $[Co(NH_3)_4](ReO_4)_2$ [95]. Complete assignment of vibrational bands, including the far-IR region, have been given in [58]. Five $\delta$(N-H) bands were observed in the IR spectrum of $[Ni(NH_3)_4](ReO_4)_2$ between 1345–1100 $cm^{-1}$ (instead of the singlet one given in [82]) and splitting of $\rho(NH_3)$ at ~680 $cm^{-1}$ was also observed together with two shoulders at 645 and 580 $cm^{-1}$. The far-IR bands show that the $ReO_4$ units are not isolated tetrahedrons but rather are joined to the complex cations to form a polymeric chain. The presence of the Ni–O–Re bond is characterized by the band observed at 219 $cm^{-1}$ [58].

The magnetic moment value ($\mu_{eff}$ = 3.08 BM for Ni complex) is temperature independent, which shows a hexacoordinated nickel(II) environment. The UV–Vis spectrum of $[Ni(NH_3)_4](ReO_4)_2$ shows tetragonal distortion (10,600 $cm^{-1}$ ($^3B_{1g} \rightarrow ^3B_{2g}$ or $^3B_{1g} \rightarrow ^3E_g$), 16,100 $cm^{-1}$ ($^3B_{1g} \rightarrow ^3A_{2g}$), 21,300 $cm^{-1}$ (no assignation), 27,000 and 25,600 $cm^{-1}$ ($^3B_{1g} \rightarrow ^3E_g$ or $^3B_{1g} \rightarrow ^3A_{2g}$), although the band assignments due to overlapping is hard in the case of similar distorted low-symmetry structures [82].

The ground state of $Co^{II}$ ion is $^4F$, and the next quartet-state (4P) is located at 11 kK higher energy. The ground term splits in the tetrahedral field ($^4A_2$ (ground state), $^4T_2$ and $^4T_1$), and according to this, the possible transitions are $^4A_2 \rightarrow ^4T_2$, $^4A_2 \rightarrow ^4T_1$(F) and $^4A_2 \rightarrow ^4T_1$(P). The first transition cannot be seen, but the other two were clearly observed at 10.0 and 18.5 kK [95]. Detailed evaluation of the electronic spectrum shows a tetrahedral Co(II) environment and weak Co-N dative bond in this compound, confirmed by the temperature-dependent magnetic measurements ($\mu_{eff}$ = 4.50–4.54 B.M. between 88 and 306 K) [82].

[Tetraamminenickel(II)] perrhenate has a phase transition at 188 K with 0.307 kJ/mol enthalpy change. The low hysteresis and entropy values found by the DSC measurement suggest a one-mechanism second-order phase transition. The stepwise decomposition of [tetraamminenickel(II)] perrhenate was followed by thermal analysis methods [82]. Non-isotherm heating between 160–195 °C [95]/141–210 °C [58]) resulted in the loss of two ammonia molecules and the formation of $[Ni(NH_3)_2](ReO_4)_2$ [58,82]. The activation energy of the decomposition step was found to be 100.62 and 98.52 kJ/mol with Kissinger–Akahira–Sunose and Kissinger methods, respectively [58].

[Tetraamminecobalt(II)] perrhenate decomposes on strong heating in air resulting in cobalt oxide as the final product [95,100]. The thermal decomposition shows three separated endotherm steps, with 2 ammonia and $Re_2O_7$ loss in the first two steps each and in the last step, at 130–190, 190–250, and 600–800 °C range, respectively [95].

### 2.4.5. $[Co(NH_3)_4CO_3]MnO_4$

[Carbonatotetraamminecobalt(II)] permanganate is a key precursor for the preparation of Co-Mn spinel oxide nanocomposite used as a catalyst in Fischer–Tropsch synthesis [19,20]. Addition of one equivalent of solid $KMnO_4$ to an aqueous solution of $[Co(NH_3)_4CO_3]NO_3$ resulted in the formation of a precipitate of $[Co(NH_3)_4CO_3]MnO_4$ in 82% yield [19,20]. The IR spectrum of $[Co(NH_3)_4CO_3]MnO_4$ contains bands at 3301 and 1621 cm$^{-1}$ that belong to the stretching frequencies of the $\upsilon$(N-H) and $\upsilon$(C=O) modes of the coordinated ammonia and carbonate ligands, respectively. The band located at 899 cm$^{-1}$ is assigned to the antisymmetric $\nu_3$(Mn–O) stretching mode. The powder X-ray diffractogram and TG/DSC curves of $[Co(NH_3)_4CO_3]MnO_4$ have been reported. It decomposes at 250 °C into Co-Mn-oxides. The TGA data of $[Co(NH_3)_4CO_3]MnO_4$ show the existence of two thermal decomposition steps, the first around 130 °C (removal of water) (The authors declare the leaving of physisorbed and crystallization water; however, the compound was given as anhydrous) and the second stage at 150–210 °C belongs to disruption of the complex structure. No weight loss was observed above 250 °C. The DSC curve shows one endothermic peak between 80 and 130 °C, whereas the decomposition has two exothermic effects between 130 and 250 °C [20].

### 2.4.6. [Tetraamminemetal(II)] Permanganates, Pertechnetates, and Perrhenates of Platinum Group Metals, $[M(NH_3)_4](XO_4)_2$ (M = Pt, Pd, X= Mn, Tc, Re) and $[Ru(NO)(OH)(NH_3)_4](ReO_4)_2$

[Tetraamminepalladium(II)] permanganate and perrhenate or [tetraammineplatinum(II)] pertechnetate were first prepared by mixing ice-cooled concentrated aqueous solutions of $[M(NH_3)_4]Cl_2$ (M = Pd, Pt) complexes and an aqueous solution of a stoichiometric amount of $NaMnO_4$ (Pd) or $NaReO_4$ (Pd, Pt). After slow evaporation in air, needle-shaped single crystals were obtained in 75–80% yield [90,93]. [Tetraammineplatinum(II)] perrhenate was also synthesized in 80% yield by boiling solid silver perrhenate with an aqueous solution of $[Pt(NH_3)_4]Cl_2$ for 40 min [93]. The analogous [tetraammineplatinum(II)] pertechnetate(VII) was synthesized as colourless platelet single crystals by adding two equivalents of $NH_4TcO_4$ in the aqueous solution of $[Pt(NH_3)_4]Cl_2$ and keeping it at room temperature for 2–4 days [91].

The [tetraamminepalladium(II)] permanganate and perrhenate are colorless triclinic crystals, but they are not isomorphic (Table 1) [90]. The monoclinic polymorph of [tetraammineplatinum(II)] perrhenate consists of colorless, non-hygroscopic plates, which are isometric in the plan, but there are also prismatic elongated varieties. The double-sided crystals exhibit oblique extinction towards the prismatic faces and the pinacoids. The optical sign is minus, and there is no pleochroism. There is no refractive index dispersion, $N_g = 1.715$, $N_m = 1.714$, $N_p = 1.676$ [93]. This polymorph is slightly soluble in water [93]. The isomorphic triclinic compounds (Pd and Re, or Pt and Tc or Re) consist of two tetrahedral $XO_4^-$ anions and a square $[M(NH_3)_4]^{2+}$ cations, linked by Re-O$\cdots$H–N hydrogen bonds. The polyhedral complex cations form hexagonal layers in the *y-z* plane. Every Pd or Pt atom is surrounded by twelve Re atoms giving hexagonal prisms [91,93]. The IR spectrum of $[Pt(NH_3)_4](ReO_4)_2$ did not show a specific influence on the energetical states of complex constituents, although the IR forbidden symmetric stretching ($\nu_1$) mode of perrhenate ion can be seen at 971 cm$^{-1}$ [93].

The thermal decomposition of $[Pd(NH_3)_4](MnO_4)_2$ was studied both in $H_2$ and He atmospheres. A thermal explosion occurred at ~200 °C; the products were X-ray amorphous, and annealing at 200–400 °C in an inert or reducing atmosphere did not improve their crystallinity [90]. The thermolysis of $[Pd(NH_3)_4](ReO_4)_2$ in helium atmosphere started at 210 °C according to the following equation:

$$[Pd(NH_3)_4](ReO_4)_2 = Pd + 2NH_4ReO_4 + 4/3NH_3\uparrow + 1/3N_2\uparrow$$

The next decomposition step corresponds to the $NH_4ReO_4$ decomposition resulting in a mixture of metallic Pd and X-ray amorphous Re oxides [90]. The thermal decomposition in an $H_2$ atmosphere at temperatures above 300 °C resulted in the formation of single-

phase powder with a hexagonal lattice. This product is a solid solution based on a rhenium structure with the composition $Pd_{0.33}Re_{0.67}$ [90].

[Tetraammineplatinum(II)] perrhenate decomposition starts at 370 °C in air and ends at 444 °C. The decomposition residue is metallic Pt, and the rhenium is released as $Re_2O_7$ [93].

$$2 [Pt(NH_3)_4](ReO_4)_2 + 5O_2 = 2Pt + 2Re_2O_7 + 4N_2 + 12H_2O$$

In a hydrogen atmosphere, the decomposition starts even at 200 °C with the formation of $NH_4ReO_4$ and finely dispersed Pt in 45 min. Further annealing of the decomposition residue in $H_2$ at 250 °C for 3 h gave $ReO_3$ and Re in addition to $NH_4ReO_4$ and Pt, whereas after 5 h annealing, only Pt and Re metal phases could be found [92]. If $[Pt(NH_3)_4](ReO_4)_2$ was heated under hydrogen at 200 °C for 45 min, then subsequently at 600 °C for 3 h, $Pt_{0.35}Re_{0.65}$ monophase solid solution was formed. A two-phase $ReO_3$-containing Pt-Re alloy was formed at 700 °C in 3 h due to the oxidation of Re by air, which contacted the sample during taking out from the furnace. On heating of $[Pt(NH_3)_4](ReO_4)_2$ at 900 °C in a hydrogen atmosphere for 7 h, the product was a $Pt_{0.33}Re_{0.67}$ solid solution (alloy) [92].

$[Ru(NO)(OH)(NH_3)_4](ReO_4)_2$ was prepared from $[Ru(NO)(OH)(NH_3)_4]Cl_2$ and $NH_4ReO_4$. It is orthorhombic, space group Pbca, Z = 8. The NO ligand is trans to the hydroxide-ion ligand, and the $ReO_4^-$ anion is H-bonded to the complex cation [112].

### 2.5. Pentaammine Complexes

One permanganate compound, [(chlorido)pentaaminecobalt(III)] permanganate, and seven perrhenate complexes, [(aquo)(pentaammine)cobalt(III)] perrhenate, $[M(NH_3)_5Cl](ReO_4)_2$ (M = Co, Cr, Ru, Rh, and Ir) complexes, and $[Pt(NH_3)_5Cl](ReO_4)_3$ are known so far. Four coordinated perrhenate containing complexes of cobalt(III), $[Co(NH_3)_5ReO_4]X_2$ (X = $ReO_4$, $ClO_4$, Cl and $NO_3$) were also described. The crystallographic data of $[M(NH_3)_5Y](XO_4)_n$ compounds are given in Table 2. All known $[M(NH_3)_5Cl](ReO_4)_2$ complexes (X = Co, Cr, Ir, Rh, Ru, Co) are isomorphic with each other [25] and with the orthorhombic $[Co(NH_3)_5Cl](MnO_4)_2$ [62] (Table 2). The incorporation of two crystalline water into the structure of [chlorido(pentaammine)cobalt(III)] perrhenate did not change the crystal structure (orthorhombic, $Cmc2_1$, Z = 8), whereas the hemihydrate of this salt has monoclinic crystals [113] (Table 2).

**Table 2.** Crystallographic parameters of $[Co(NH_3)_5Y](XO_4)_2$ compounds.

| Compound | T, K | a, b, c | α, β, γ | Space Group | Z | V, Å³ | $D_{calcd}$, g/mL | Ref. |
|---|---|---|---|---|---|---|---|---|
| M = Co, Y = H₂O, X = Re, n = 3, ×2H₂O | 150 | 9.9797 12.6994 14.7415 | 102.870 | C2/c | 4 | 1821.35 | 3.456 | [113] |
| M = Co, Y = Cl, X = Re, n = 2, ×2H₂O | | 14.9446 14.6562 12.2434 | | $Cmc2_1$ | 8 | 2681.68 | 3.368 | [113] |
| M = Co, Y = Cl, X = Re, n = 2; ×0.5H₂O | 293 | 8.0086 12.9839 17.5122 | 91.858 | $P2_1/sn$ | 4 | 1820.01 | 3.462 | [113] |
| M = Co, Y = Cl, X = Re, n = 2 | 293 | 14.974 14.688 12.2434 | | | | 2708.5 | 3.33 | [25] |
| M = Rh, Y = Cl, X = Re, n = 2 | 293 | 15.0740 14.7290 12.3470 | | $Cmc2_1$ | | 1430.5 | 3.19 | [25] |

**Table 2.** *Cont.*

| Compound | *T*, K | *a, b, c* | *α, β, γ* | Space Group | *Z* | *V*, Å³ | $D_{calcd.}$, g/mL | Ref. |
|---|---|---|---|---|---|---|---|---|
| M = Cr, Y = Cl, X = Re, n = 2 | 293 | 15.071 14.806 12.439 | | | | 2775.7 | 3.30 | [25] |
| M = Ru, Y = Cl, X = Re, n = 2 | 293 | 15.053 14.793 12.445 | | | | 2741.3 | 3.54 | [25] |
| M = Rh, Y = Cl, X = Re, n = 2 | 293 | 15.033 14.723 12.331 | | | | 2729.2 | 3.55 | [25] |
| M = Ir, Y = Cl, X = Re, n = 2 | 293 | 15.059 14.718 12.359 | | | | 2739.2 | 3.98 | [25] |
| M = Co, Y = Cl, X = Mn, n = 2 | | 14.2753 14.2816 12.2342 | | Cmc2₁ | 8 | 2494.24 | 2.216 | [62] |
| M = Pt, X = Cl, X = Re, n = 3, ×2H₂O | | 10.3476 12.8955 14.3536 | | P2₁/n | 4 | 1847.94 | 3.962 | [114] |

### 2.5.1. [(Chlorido)pentamminecobalt(III)] Permanganate

[Chlorido(pentaammine)cobalt(III)] permanganate was prepared first by Franguelli et al. [62] by the reaction of $[Co(NH_3)_5Cl]Cl_2$ and excess of concentrated $NaMnO_4$, as an orthorhombic (Table 1) reddish-purple crystalline mass with a 57% yield [62]. Previously, Krestov and Yatsimirksii calculated its theoretical thermodynamical properties [115]. Its solubility in water is 3.71 g/L and 9.39 g/L at 0 °C and 25 °C, respectively. It is insoluble in organic solvents like aliphatic hydrocarbons, chloroform, dichloromethane, or benzene and soluble in DMF but decomposes in DMSO [62].

The $Co^{3+}$ cation is surrounded by five ammonia and one coordinated chloride ion in a distorted octahedral arrangement. The asymmetric unit contains two half cations and two permanganate anions (Figure 4). The cobalt ion, the chloride ion, and a part of ammonia ligands sit on a mirror plane (have half-site occupancy); thus, the hydrogens of these ammonia ligands are disordered over two mirrored positions. The axial Co-N bond distance is shorter and longer than that of the equatorial ones in cation B and cation A, respectively. The Co-Cl bonds were almost equal in cations A and B. Layer of the cations in the bc plane was embedded between permanganate layers [62].

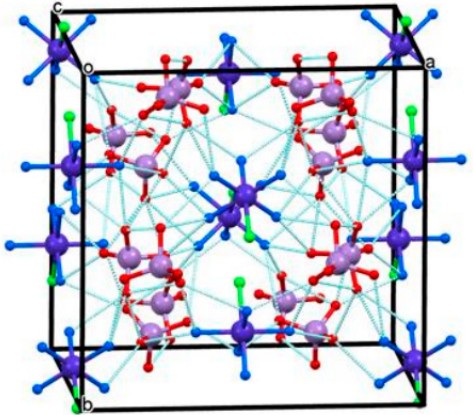

**Figure 4.** Packing arrangement of $[Co(NH_3)_5Cl](MnO_4)_2$ and hydrogen bonds in light blue (hanging contacts are omitted for clarity) between the ions. Reproduced from [62].

The $[Co(NH_3)_5Cl]^{2+}$ cation in $[Co(NH_3)_5Cl](MnO_4)_2$ has five hydrogen donor groups with $3 \times 5$ donor sites and one hydrogen acceptor group (chloride-ion). The cations are arranged in the crystal lattice in such a way that all ammonia hydrogens can form hydrogen bonds with at least one acceptor (chloride or permanganate oxygen) atom [62]. Detailed spectroscopic analysis (correlation analysis, low-temperature IR, Raman, and far-IR) spectroscopic studies have been carried out, and all vibrational bands appearing in the IR and Raman spectra of $[Co(NH_3)_5Cl](MnO_4)_2$ have been assigned [62]. The singlet nature symmetric stretching mode of permanganate ion appears as a twin peak in the low-temperature Raman spectra, which confirms the presence of two crystallographically and spectroscopically different permanganate ions [62]. The $\rho(NH_3)$ rocking mode is sensitive to the presence of hydrogen bonds in ammonia complexes, but the $\rho(NH_3)$ IR band of $[Co(NH_3)_5Cl](MnO_4)_2$ is coinciding with its $\nu_s(Mn-O)$ mode. Therefore, deuteration experiments were carried out to assign the $\rho(ND_3)$ band position (646 cm$^{-1}$), and with the use of the $\rho(NH_3)/\rho(ND_3)$ wavenumber ratio found for $[Co(NH_3)_5X]X_2$ (X = halogen) compounds [116], the position of $\rho(NH_3)$ could be calculated (818 cm$^{-1}$). It showed that the strength of the hydrogen bonds in the structure of $[Co(NH_3)_5Cl](MnO_4)_2$ lies between that of in $[Co(NH_3)_5Br]Br_2$ ($\rho(NH_3)$ = 830 cm$^{-1}$) and $[Co(NH_3)_5I]I_2$ ($\rho(NH_3)$ = 810 cm$^{-1}$) [62,116].

No important differences were found between the thermal decomposition characteristics of [(chlorido)pentaamminecobalt(III)] permanganate in argon and air atmospheres. The decomposition product formed above 400 °C was $CoMn_2O_4$ (average crystallite size of 16.8 nm) [62]. The thermal decomposition of [(chlorido)pentaamminecobalt(III)] permanganate starts at 121 °C. The ammonia molecules are bonded with various strengths ($\varepsilon$ = 0.80 and 0.88 [117], which predicted stepwise ammonia loss. The TG-MS study showed that the releasing material in the first decomposition step is not ammonia. Either $N_2$ or $NO_x$ compounds were not formed in this decomposition step. Since [(chlorido)pentaamminecobalt(III)] permanganate is anhydrous, and the decomposition was conducted in an inert atmosphere, and water as the reaction product could be formed in the first step only as a product of a solid-phase redox reaction between ammonia ligand and permanganate ion (the only hydrogen and oxygen sources are ammonia and permanganate ion, respectively). As a result of this, no permanganate ion was found in the amorphous reaction product of the first decomposition step [62].

The decomposition was performed in a refluxing toluene solvent since, in this case, the decomposition temperature could not be exceeded the boiling point of toluene. Furthermore, the liberated reaction heat was turned to evaporate the toluene. The amorphous product, after heating, was washed out with water when the soluble phases were identified as $NH_4NO_3$, $[Co(NH_3)_5Cl]Cl_2$, and $NH_4Cl$. The solid residue based on the elementary analysis and material balance was a cobalt manganese oxide containing ammonia and formed according to the following equation [62]:

$$3[Co(NH_3)_5Cl](MnO_4)_2 = [Co(NH_3)_5Cl]Cl_2 + 3H_2O + 3NH_4NO_3 + \{Co_2(NH_3)_4Mn_6O_{12}\}$$

Further studies showed that the ammonia content is non-coordinated, but it is in ammonium-ion form, and the real formula of the amorphous solid can be written as $[(NH_4)_4Co_2Mn_6O_{12}]$ [61]. This composition is very similar to the todorokite mineral group members. The water-soluble and insoluble components of the amorphous decomposition product mixture react with each other on heating above 400 °C, which resulted in the formation of $CoMn_2O_4$ and gaseous ($N_2$, $H_2O$, $NH_3$, $NH_4Cl$) products. The crystallite size and photocatalytic activity of the stoichiometric $CoMn_2O_4$ products in the degradation of organic dyes strongly depended on the heating time and temperature [62]. Heating of the amorphous $(NH_4)_4Co_2Mn_6O_{16}$ also gave a spinel-like decomposition product, but with Co:Mn = 1:3 stoichiometry [62].

2.5.2. [Pentamminechlorometal(III)] Perrhenates, $[M(NH_3)_5Cl](ReO_4)_2$ (M = Co, Cr, Ru, Rh, Ir)

Stirring of dilute solutions of [chloropentaamminemetal(III)] dichlorides ($[M(NH_3)_5Cl]Cl_2$, M = Co, Cr, Ru, Rh, Ir, 0.01 M) and $NaReO_4$ (0.01 M) resulted in an almost immediate formation of a needle-like pink (M = Co), light brown (M = Cr), dark yellow (M = Ru) or light yellow (M = Rh, Ir) crystalline precipitate in ~80% yield [25,113]. The single crystals of the cobalt complex hemihydrate were obtained by interdiffusion for a few days in agarose gel in a U-tube at room temperature (one end of the tube was filled with a diluted aqueous solution of $[Co(NH_3)_5Cl]Cl_2$ and another one was filled with two equivalents of $NaReO_4$) [113]. The single crystals of the Rh complex were obtained by slowly evaporating a dilute aqueous solution of $[Rh(NH_3)_5Cl]Cl_2$ and two equiv. of $NaReO_4$ at room temperature [25]. The lattice parameters of [pentaamminechlorometal(III)] perrhenates, $[M(NH_3)_5Cl](ReO_4)_2$ (M = Co, Cr, Ru, Rh, Ir) complexes and $[Co(NH_3)_5Cl](MnO_4)_2$ are given in Table 2.

These complexes are anhydrous, except the Co complex, which has hemi and dihydrate forms as well. The dihydrate consisting of distorted octahedral $[Co(NH_3)_5Cl]^{2+}$ cations occupy two crystallographically independent sites oriented to symmetry elements in a different way, and there are two crystallographically independent perrhenate anions [113]. The cationic and anionic layers along the x-axis are bound inside and between themselves by weak hydrogen bonds (N–H . . . O-Re and N–H . . . Cl). Water molecules are located in the cation layers: the minimum Co . . . Co distance is 5.968 Å in the cation layer; the shortest $O_w$ . . . N-H contact is 3.02 Å. In the perrhenate anion layer, the shortest Re . . . Re distances are 4.224–4.457 Å [113].

The structure of the rhodium complex (and the other isomorphic ones) is an island type and consists of $[Rh(NH_3)_5Cl]^{2+}$ octahedra and $ReO_4$-tetrahedra with two crystallographically independent cationic and two anionic units. The one kind (1) of rhodium ions lie on a mirror plane; the Cl and one of the ammonia N atoms from the Rh environment are in the m plane, whereas other N atoms are pair wise related by this plane; the Cl and three N atoms from the other (2) rhodium environment lie in the m plane, and two N atoms are related by this plane [25]. Each $[Rh(NH_3)_5Cl]^{2+}$ cation is surrounded by twelve perrhenate ions forming a distorted hexahedral prism. The Rh . . . Rh distances in the distorted prism lie between 4.921 and 6.391 Å; the Re . . . Re distances are between 4.25 and 4.48 Å. The second coordination sphere of the perrhenate anion involves six cations with Rh . . . Re distances between 5.372 and 6.336 Å. Along the y axis layers of $[Rh(NH_3)_5Cl]^{2+}$, cations and perrhenate anions with intra- and interlayer hydrogen bonds (N–H . . . O-Re and N–H . . . Cl of 2.91–3.29 Å and 3.33–3.53 Å, respectively) are observed. The shortest interionic contacts Re-O . . . O-Re and Re-O . . . Cl are 3.10 Å and 3.36 Å [25].

The thermal decomposition of $[M(NH_3)_5Cl](ReO_4)_2$ (M = Co, Cr, Ru, Rh, Ir) in a helium atmosphere resulted in coinciding decomposition steps. The decomposition products are amorphous/multiphase oxide materials. In a hydrogen atmosphere at 600 °C, however, the thermal decomposition product is a monophase binary alloy, $M_{0.33}Re_{0.67}$, (M = Co, Ru, Ir, Rh), which are ultrafine metal powders (solid solutions) derived from hexagonal close packing of rhenium [25]. In the case of the chromium complex, in a hydrogen atmosphere, multiphase decomposition products are formed consisting of only one crystalline compound characterized by the $Cr_{0.003}Re_{0.997}$ formula [25].

The rhodium and iridium complexes decompose in an analogous way. The thermal decomposition of [chloropentaamminerhodium(III)] perrhenate in a step-by-step quenching under an $H_2$ atmosphere showed that at 190 °C (first step), both the cation and anion are partially reduced, resulting in appearing of a very broad diffuse maximum at the angles corresponding to the most intense peaks of the metallic rhenium (100, 002, 101) and rhodium (111, 200). The reflections of the $[Rh(NH_3)_5Cl](ReO_4)_2$ have completely vanished at 200 °C and the peaks belonged to $NH_4ReO_4$, $NH_4Cl$ and $[Rh(NH_3)_5Cl]Cl_2$ have emerged. At 220 °C, in the third stage, only a small amount of $NH_4ReO_4$ and the broad peaks of the $Rh_{0.70}Re_{0.30}$ solid solution with hcp lattice were found. It shows that the metallic solid solution formed at the first stage is definitely rhenium-deficient. At the

fourth stage, between 220 and 280 °C, the $NH_4ReO_4$ content was completely decomposed, and the formed rhenium penetrated into the lattice of the solid solution resulting in a gradual increase in the lattice parameters corresponding to the composition $Rh_{0.30}Re_{0.70}$. The vacuum annealing of this solid solution at 950 °C for 400 h leads to an increase in crystallite sizes [26].

The in situ synchrotron X-ray studies on thermal decomposition products under an $H_2$ atmosphere showed a very wide band appearing at the decomposition stage at ~200 °C, whereas further heating resulted in the formation of the solid solution with $Rh_{0.40}Re_{0.60}$ composition with hcp lattice of rhenium [27]. Further raising of temperature up to 340 °C was accompanied by an extension of the unit cell dimensions and the formation of the solid solution with $Rh_{0.33}Re_{0.67}$ composition. In the first stage of the decomposition reaction, metallic rhodium develops as an amorphous non-diffracting phase, whereas rhenium is formed in the second stage of the process and penetrates into nanosized rhodium particles. The $Rh_{0.40}Re_{0.60}$, $Rh_{0.35}Re_{0.65}$, and $Rh_{0.34}Rh_{0.66}$ alloys were formed at 206, 230, and 250 °C, respectively, whereas above 280 °C, only the final product, $Rh_{0.33}Re_{0.67}$, could be detected [27]. Contrary to the experiments carried out by step-by-step quenching of the reaction products, no $Rh_{0.30}Re_{0.70}$ and other intermediates such as $NH_4ReO_4$, $NH_4Cl$, and $[Rh(NH_3)_5Cl]Cl_2$ were observed [27].

### 2.5.3. [Pentammine(aquo)cobalt(III)] Perrhenate Dihydrate

The addition of a 10% ammonia solution to an aqueous solution of $[Co(NH_3)_5Cl]Cl_2$ followed by an aqueous solution of $NaReO_4$ resulted in a complete precipitation of $[Co(NH_3)_5H_2O](ReO_4)_3 \cdot 2H_2O$. The yield was 90%. Its single crystals were grown in 3 days in a U-tube filled with a diluted aqueous $[Co(NH_3)_5Cl]Cl_2$ and two equiv. of $NaReO_4$ (0.41 mmol) [113]. With the use of a 5.4 M $HReO_4$ solution and an aqueous solution of $[Co(NH_3)_5OH_2](ClO_4)_3$, the pink $[Co(NH_3)_5(H_2O)](ReO_4)_3 \cdot 2H_2O$ was formed in 30 min in >95% yield [114]. A $^{18}OReO_3$ labeled sample was prepared in an analogous way from $NaReO^*_4$ isotope labeled perrhenate salt. Its dehydration at 50–60° and at 0.001 Hgmm vacuum for 3 h resulted in complete loss of the lattice water and the formation of pink $[Co(NH_3)_5(H_2O)](ReO_4)_3$ [118]. No isotope exchange was observed between the perrhenate ion and crystallization water during the dehydration process [118].

The complex cation in $[Co(NH_3)_5H_2O](ReO_4)_3 \cdot 2H_2O$ has a slightly distorted octahedral structure. The cis angle deviations are not much than 1.9°. The length of the Co–N bond located in the trans-position towards water is shorter only on average by 0.03 Å than the other bonds [113]. The structural units are hydrogen bonded involving the crystalline water molecules. The intermolecular interactions in the $O_w$ ... O-Re linkages are stronger than in the $O_w$ ... N-H linkages. Each cation is surrounded by 12 perrhenate anions; the Co ... Re distances between the centers are 5.297–5.875 Å; the shortest Re ... Re distances in the perrhenate layers are 4.307–4.885 Å [113]. The IR spectrum in the perrhenate normal modes region was also recorded and analyzed [118].

### 2.5.4. [Perrhenatopentamminecobalt(III)] Perrhenate, $[Co(NH_3)_5OReO_3](ReO_4)_2$

Dehydration of $[Co(NH_3)_5(H_2O)](ReO_4)_3$ between 95 and 130° at 0.001 mmHg pressure resulted in a quantitative loss of the coordination water in a period of 2–5 h. No ammonia evolution was detected. The solid residual product is $[Co(NH_3)_5(OReO_3)](ReO_4)_2$. This salt is purple and quite insoluble in water [118]. The $^{18}O$-perrhenate labeled compound was treated in a vacuum at 110–120 °C when only 2% of isotope exchange was observed between the perrhenate ion and the coordinated water [118]. The aquation of $[(NH_3)_5Co-OReO_3]^{2+}$ ion in the aqueous solution of $[Co(NH_3)_5OReO_3](ReO_4)_2$ and on the fixed form of ion-exchange resins were followed at different pH values. The $O^{18}$ isotopic exchange of $H_2O$ with $[(NH_3)_5Co(H_2O)]^{3+}$ and $ReO_4^-$ were also studied. The aquation of $[(NH_3)_5CoOReO_3]^{2+}$ ($H_2O$-$ReO_4^-$ exchange) was faster in the acidic region and slower in the basic region. A new term of the rate equation was introduced when $ReO_4^-$ was on the resin [119]. The IR spectra in the perrhenate normal modes region were recorded [118].

### 2.5.5. $[Co(NH_3)_5ReO_4]X_2$ (X = Cl, $ClO_4$, $NO_3$)

A Dowex 1-X4 ion exchanger resin was converted to its chloride, nitrate, or perchlorate form and washed with 0.001 M 2,6-lutidine (to retard hydrolysis of Co complex) solution, then the slightly soluble $[Co(NH_3)_5OReO_3](ReO_4)_2$ was ground with the resin and water at 10–15°, and the solution was separated. The addition of saturated LiCl, $LiNO_3$, and $LiClO_4$ solutions at 0 °C resulted in crystalline $[Co(NH_3)_5OReO_3]X_2$ compounds in 60–80% yield. The violet-red chloride salt is not stable for a long time, even in the solid state, and an internal replacement takes place with the chloride ion resulting in some $[Co(NH_3)_5Cl]^{2+}$ derivative [118–120]. The violet-red perchlorate salt, however, is stable in the solid state over a period of 6 months [114–116]. Both salts are soluble in water and hydrolyze with the formation of $[Co(NH_3)_5(H_2O)]^{2+}$ solution with >95% yield. The kinetics of these reactions in acidic, alkaline, and neutral solutions were analyzed in detail. The first-order rate constant, the kinetic parameters, and their temperature dependences are given. They show an absorption maximum of 530 nm [118]. The hydrolysis kinetics of the nitrate complex was studied in more detail using an $^{18}O$-perrhenate-containing sample, which was prepared in an analogous way as the non-labeled nitrate salt. The first-order rate is constant, and the kinetic parameters and their temperature dependence are given, including the results of acetic acid/acetate ion catalysis in the hydrolysis reaction between pH= 4 and 7, which proceeds with 80–95% conversion. It has an absorption maximum of 530 nm [114]. The isotope-labeled $[Co(NH_3)_5OReO_3](NO_3)_2$ was dissolved in water at the appropriate pH and buffered with 2,6-lutidine and $HNO_3$. The isotope exchange between the isotope-labeled perrhenate-ion and water was fast and complete [118]. The rhenium-oxygen fission is shown to be the primary process in the hydrolysis catalyzed by acids and bases and the isotope transfer experiments. The rate constants of aquation with $Co-OReO_3$ fission show that this complex aquation mechanism is different from some other analog [acido-pentaamminecobalt(III)] complexes. The dissociation at pH 4–5 occurs predominantly by Co-O fission [118].

### 2.5.6. [Pentaamminechloroplatinum(IV)] Perrhenate

$[Pt(NH_3)_5Cl](ReO_4)_3 \cdot 2H_2O$ was synthesized from an aq. solution of $[Pt(NH_3)_5Cl]Cl_3 \cdot H_2O$ and an excess of hot aq. solution of $NaReO_4$. Colorless crystals as elongated monoclinic platelets appear in 1 h; the yield was 75–80% [114]. Its structure consists of a $[Pt(NH_3)_5Cl]^{3+}$ cation with three and two crystallographically independent $ReO_4^-$ anions and crystallization water, respectively. The cations and anions are in general positions. The bond angles of the tetrahedral anion and the octahedral cation deviate from ideal values. An extended network of hydrogen bonds involving the cation, anion, and molecules of crystallization water is built up. The shortest intermolecular contacts involving water are $H_2O \ldots Cl= 3.35$ Å, $H_2O \ldots N = 2.86$ Å, $H_2O \ldots$ O-Re = 2.75 Å. The $[Pt(NH_3)_5Cl]^{3+}$ cations are surrounded by 12 $ReO_4^-$ anions, whereas the anions have four cationic neighbors and build up an almost regular tetrahedron. The packing of the ions is perpendicular to the direction [101]. The perrhenate anions form a hexagonal network, which is alternated with the hexagonal layer of $[Pt(NH_3)_5Cl]^{3+}$ cations. The crystallization water molecules occupy the tetrahedral voids within the anionic framework. The cationic sublattice approach was used to determine the M $\ldots$ M distances. The Re $\ldots$ Re distances fall within 4.38–4.80 Å, the Pt $\ldots$ Pt distances at 7.68, whereas the Pt $\ldots$ Re distances between 5.32–5.64 Å, and in general, the average distances between metal atoms correspond to the edge lengths of the distinguished rhombohedra [114].

Its decomposition in an inert atmosphere proceeded with the elimination of crystalline water at 110–140 °C, followed by a decomposition step starting at 280 and ending at 500 °C. The decomposition product consists of two solid solutions, one Pt and another Re-based one. In an $H_2$ atmosphere, a $Re_{0.75}Pt_{0.25}$ alloy was formed with Re-structure [114].

### 2.6. Hexammine Complexes

The [hexaamminecobalt(III)] and [hexaamminechromium(III)] permanganates are anhydrous isomorphic cubic crystals, whereas the perrhenate (orthorhombic) and pertech-

netate (monoclinic) salts of the [hexaamminecobalt(III)] cations are dihydrates, but are not isomorphic either with each other or the anhydrous permanganate salt [98,111,113] (Table 3).

**Table 3.** Crystallographic data of $[M(NH_3)_6](XO_4)_n$ [hexaamminemetal] permanganates, pertechnetates and perrhenates.

| Compound | T, K | a, b, c | α, β, γ | Space Group | Z | V, Å³ | $D_{calcd}$, g/mL | Ref. |
|---|---|---|---|---|---|---|---|---|
| M = Co, X = Re, n = 3, ×2H₂O | 293 | 14.9446 14.6562 12.2434 | | Cmc2₁ | 8 | 2681.68 | 3.368 | [113] |
| M = Co, X = Mn, n = 3 | | 11.39 | | $T_d^2$--F4-3m | 4 | 1477.6 | 2.33 | [111] |
| | | 11.39 | | | 4 | 1477.65 | | [121] |
| M = Co, X = Tc, n = 3, ×2H₂O | | 8.0266 12.6275 17.6438 | 91.320 | P2₁/n | | | | [98] |
| M = Cr, X = Mn, n = 3 | | 11.45 | | $T_d^2$-F4-3m | 4 | 1501.1 | 2.26 | [111] |

2.6.1. $[Ni(NH_3)_6](MnO_4)_2$

The reaction of nickel(II) sulfate solution with ammonia and saturated potassium permanganate solution below 5° resulted in a blackish small crystal deposit in twenty to thirty minutes, declared to be the dihydrate of [hexaamminenickel(II)] permanganate [96]. It is soluble without decomposition in diluted sulfuric acid, but from the aqueous solutions, which are initially beautiful purple in color, quickly manganese oxide is deposited. It detonates under the shock of the hammer, heating or rubbing/crushing in a mortar, fusing with releasing ammonia and producing a cloud of very finely divided oxides. After five to six days, it has partially decomposed. This salt was later found to be impure because, in spite of using the chilled nickel salt and potassium permanganate solutions, on adding the ammonia solution, the reaction heat caused partial decomposition. However, adding an excess of aqueous ammonia to the nickel nitrate solution and cooling the mixture with ice, and adding two equiv. of ice-cold potassium permanganate solution, [hexamminenickel(II)] permanganate that formed was found to be very pure and anhydrous [118]. It can be dried at temperatures below 100 °C or in a vacuum over caustic soda in the dark. The [hexamminenickel(II)] permanganate can also be prepared in the reaction of the [hexaamminenickel(II)] chloride and potassium permanganate under ice-cooling [111]. It consists of isotropic octahedra with violet-black color. Its aqueous solution is violet in color. In contrast to the salt described by Klobb, it is fairly stable, at least in the dark. Its complete decomposition occurs only after several months. It forms violet mixed crystals with the analog tetrafluoroborate. These mixed crystals were more stable than the pure permanganate compound—no decomposition was started even after six months of storage—and dissolved in water with purple color without leaving a residue. [122]. The $[Ni(NH_3)_6](MnO_4)_2$ cannot be indexed as cubic material, although the corresponding perchlorate is cubic. The powder X-ray data, including the d-values and intensities, have been reported [111]. The IR band assignment for the cationic and anionic parts is given and evaluated in detail. The presence of hydrogen bonds has an influence on the rocking and symmetric deformation N-H modes [111].

2.6.2. [Hexaamminenickel(II)] Perrhenate, $[Ni(NH_3)_6](ReO_4)_2$

[Hexaamminenickel(II)] perrhenate was prepared in the reaction of the [hexaamminenickel(II)] chloride and sodium perrhenate. Wilke-Dörfurt et al. prepared it from in situ synthesized nickel(II) perrhenate solution (dissolving nickel carbonate in 0.004 M perrhenic acid). Hetmanczyk et al. used basic nickel carbonate and 75–80% perrhenic acid

solution [58]. Evaporation of the solution in a water bath, then the addition of ammonia and cooling with an ice–sodium chloride mixture resulted in violet [hexaamminenickel(II)] perrhenate [68]. [Hexaamminenickel(II)] perrhenate has needle-shaped crystals with light purple color. It shows oblique extinction and is believed to be a triclinic prism. It easily decomposes with ammonia releasing [111], and on standing in air, gradually loses its ammonia and turns into a light blue material (fast at 100 °C [100]), which was identified by Wilke-Dörfurt et al. as $[Ni(NH_3)_4](ReO_4)_2$ [68]. It can be stored under an ammonia atmosphere for a long period of time without any decomposition [58]. Its solubility in aq. ammonia solution of specific gravity 0.930 at 26 °C: 33.4 g/L, its density at 25 °C is d = 3.000 g/mL, mol-vol 220.5 [68]. No phase transitions of the Ni complex could be detected [58]. The powder X-ray data, including the d-values and intensities and the IR bands assignment for the cationic and anionic parts, have been reported. The presence of hydrogen bonds has an influence on the position and splitting of the rocking and symmetric deformation N-H modes [111], which was confirmed with the complete assignment given on the basis of quantum chemical calculations [58]. The appearance of the IR inactive $v_1$(Re-O) IR band suggests distorted tetrahedral geometry of the perrhenate ion [58].

The thermal decomposition of $[Ni(NH_3)_6](ReO_4)_2$ is a four-step process, but the first thermal decomposition step at 300–368 K resulted in [tetraamminenickel(II)] perrhenate, without any redox reaction; thus, the further decomposition steps belong to the decomposition steps of [tetraamminenickel(II)], [diamminenickel(II)] and nickel(II) perrhenates, respectively [58]. The kinetic studies showed 76.66 and 78.21 kJ/mol activation energy with KAS and Kissinger methods, respectively. This indicated that only one mechanism is involved in the thermal decomposition of $[Ni(NH_3)_6](ReO_4)_2$ [58].

### 2.6.3. [Hexaamminemetal(III)] Permanganate, Pertechnetate, and Perrhenate, $[M(NH_3)_6](XO_4)_3$, (M = Cr, X = Mn, Re), M = Co, X = Tc, Re)

[Hexaamminechromium(III)] permanganate and the dihydrates of [hexaamminecobalt(III)] perrhenate and pertechnetate were prepared in the reaction of the [hexaamminemetal(III)] chloride (M = Cr, Co) and potassium permanganate, ammonium pertechnetate, and sodium perrhenate, as purple (M = Cr, X = Mn), lemon-yellow (M = Cr, X = Re), orange (M = Co, X = Tc) or yellowish orange (M = Co, X = Re) crystalline materials [98,111,113,122]. The yields of Co complexes with pertechnetate and perrhenate anions were found to be 56 and 80%, respectively.

The reaction of a warm solution of [hexaamminechromium(III)] nitrate and ~10 equivalent of potassium permanganate also resulted in violet-black octahedrons of [hexamminechromium(III)] permanganate, which is in a thin layer and at the edges transmit the light with a violet color. The analogous reaction with perrhenic acid resulted in the perrhenate dihydrate concentrating and cooling the solution. The formed precipitate was recrystallized from an aq. solution containing a small amount of perrhenic acid. [68], with subsequent drying over calcium chloride. When the crystals were dried for several hours over $P_2O_5$ in a vacuum, the anhydrous perrhenate salt was formed [68].

The needle-like of $[Co(NH_3)_6](ReO_4)_3 \cdot 2H_2O$ appeared immediately, and its single crystals were grown by interdiffusion in agarose gel in a U-tube at room temperature (one end of the tube was filled with a diluted aqueous solution of $[Co(NH_3)_6]Cl_3$ and another one was filled with two equivalents of $NaReO_4$ [113]. If a concentrated aq. solution of [hexaamminecobalt(III)] chloride was reacted with an excess of an equally concentrated aqueous solution of perrhenic acid, and the solution was concentrated and cooled, $[Co(NH_3)_6](ReO_4)_3 \cdot 2H_2O$ was crystallized out. The recrystallization from the water was performed in the presence of a small amount of perrhenic acid [68]. The crystals dried over calcium chloride proved to be dihydrate, which transforms into anhydrous salt in several hours over $P_2O_5$ in a vacuum [68], as that was found for the analog $[Cr(NH_3)_6](ReO_4)_3 \cdot 2H_2O / [Cr(NH_3)_6](ReO_4)_3$ pair as well.

Among the chromium complexes, the permanganate salt is sparingly soluble in water and forms mixed crystals with the perchlorate and the tetrafluoroborate. On heating, it deflagrates violently, emitting a puff of dark smoke [122] and explodes under rubbing

111]. Its pycnometric density was found to be 2.21 g/mL. The hexaamminechromium(III) perrhenate dihydrate is lemon-yellow, its small prisms showing straight eruptions [68,111]. Its solubility in water at 20 °C is 0.684 g/L, and the density and the molar volume are 3.280 g/mL and 287.5 cm$^3$/mol, respectively, whereas the same values for the anhydrous salt are 3.408 g/mL and 265.5 cm$^3$/mol [68]. The perrhenate does not belong to the regular system and is less soluble than other similar salts with tetrahedral anions (perchlorate, permanganate, fluorosulfonate, tetrafluoroborate) [68].

The powder X-ray data, including d-values and Miller indexes and the IR bands assignment for the cationic and anionic parts for both Cr complexes, were given and evaluated in detail [111].

The ammonia complex of cobalt(III) pertechnetate can be kept in air for months without decomposition. Its solubility in water at room temperature is $6.15 \times 10^{-4}$ M. The analogous cobalt perrhenate complex consists of small prisms, which show straight extinction as birefringent crystals [68,111]. Its powder X-ray data, including the d-values and intensities, are given [111]. Its solubility in water at 20 °C was found to be 0.469 g/L, and the density and molar volume is 3.506 g/mL and 260 cm$^3$/mol, whereas the same values for the dihydrate are 3.329 g/mL and 285 cm$^3$/mol [68], respectively. It does not form crystals belonging to the regular system, and its solubility is lower than that of some similar salt with tetrahedral anions (perchlorate, permanganate, fluorosulfonate, tetrafluoroborate) [68].

Monoclinic single crystals of [hexaamminecobalt(III)] pertechnetate salt were grown from a 1:1 water:methanol mixture at −15 °C after keeping for a week [98]. The asymmetric unit of the pertechnetate salt consists of three crystallographically independent $TcO_4^-$ anions, one [Co(NH$_3$)$_6$]$^{3+}$ cation, and two water molecules. The $TcO_4^-$ tetrahedra (C$_{3v}$ symmetry) and the octahedral cation are slightly distorted [98]. [Co(NH$_3$)$_6$][TcO$_4$]$_3$ 2H$_2$O exhibits a layered structure. No van der Waals interactions within the cationic layer (the shortest Co-Co and N-N distances between two distinct cationic units are 7.797 and 4.091 Å, respectively). The cationic and anionic layers are bound by hydrogen bonds; their thicknesses are 4.5 and 4.41 Å, respectively. Each cation is connected to twelve pertechnetate anions, and the shortest Co-N-H . . . .O-Tc distances vary between 2.031 and 2.089 Å. The shortest Tc-Tc separation in the anionic layer is 4.326 Å, and the pertechnetate anions are connected with each other via hydrogen bonds mediated by water molecules. The shortest Tc-O . . . H-O-H distances are 1.948 and 2.085 Å [98].

The centrosymmetric cobalt cation in the [hexaamminecobalt(III)] perrhenate coordinates six ammonia molecules, forming a practically regular octahedron. The average Co–N distances are 1.963 Å; bond angle deviations from 90° at the Co atom do not exceed 1.8° [113]. It contains two crystallographically independent perrhenate anions, the Re atom in one of them is located on the twofold axis [113]. It has layered packing having hydrogen bond interactions. The water molecules are bound with perrhenate anions via O–H . . . O-Re bonds. In the anion layers, the Re . . . Re distances between the nearest perrhenate anions are 4.357–5.949 Å; the minimum Co . . . Co distance in the cation layer was found to be 7.927 Å. The cations and anions are also bound by weak hydrogen bonds of the N–H . . . O-Re type. The shortest distances between the centers of the complex ions and anions are Co . . . Re = 5.394–5.651 Å [113].

The IR band assignments for the cationic and anionic parts for all four salts were given and evaluated in detail. The presence of hydrogen bonds has an important influence on the rocking and symmetric deformation N-H modes of these complexes [100,122].

### 2.6.4. [Hexaammincobalt(III)] Permanganate, [Co(NH$_3$)$_6$](MnO$_4$)$_3$

The [hexaamminecobalt(III)] permanganate was prepared in the reaction of the [hexaamminecobalt(III)] chloride and potassium permanganate [69,111,121,123]. The interaction of hot concentrated solutions of [hexaamminecobalt(III)] chloride with two equiv. of potassium permanganate at a temperature that must not exceed 60 °C resulted in a mixture of the almost insoluble [hexaamminecobalt(III)] permanganate and a by-product consisting of hexagonal plates on cooling. The by-product is a compound formed in higher

proportion if potassium permanganate is not taken in a large excess. The cold water dissolves the hexagonal salt well, and that can be washed out. Recrystallization from water at 60 ° resulted in black octahedra [69,119]. The use of high excess potassium permanganate at 55 °C resulted in a product of 98% purity [121]. It is a purple crystalline material that explodes under rubbing [111]. $[Co(NH_3)_6](MnO_4)_3$ forms brilliant crystals, which are very sparingly soluble in cold water (1 part requires 1388 parts of water at 0 °C) and more soluble in hot water but partially decomposes. It explodes quite strongly upon heating and under the impact of the hammer as well. It turns into manganese(II) chloride and luteocobaltic chloride with treatment with concentrated hydrochloric acid [69,123].

It is isostructural with the analog [hexaamminecobalt(III)] perchlorate as well [112], crystallizes in a cubic face-centered lattice, and its pycnometric density was given as 2.26 g/mL [112] or 2.16 g/mL [121]. Its powder X-ray diffractogram could be indexed according to a cubic cell [111].

All N-H and permanganate bands were unambiguously identified in the IR spectrum of $[Co(NH_3)_6](MnO_4)_3$. The two bands of antisymmetric Mn-O stretching mode were at 913 and 897 $cm^{-1}$. The two bands may belong to the two components of the triple degenerate $\nu_3$ modes or may belong to two kinds of permanganate ions that are located in two different positions [121]. The IR band assignment for the cationic and anionic parts and the influence of the presence of hydrogen bonds in rocking and symmetric deformation N-H modes were also evaluated [113].

The thermal decomposition of [hexaamminechromium(III)] permanganate starts with a very faint exothermic peak at 100 °C, followed by an endothermic one at 103 °C (probably the physisorbed water) and a further exothermic peak at 107 °C, and from 108 °C, it begins to decompose very quickly, which ends explosively at 116 °C [121].

2.6.5. [Hexaamminecobalt(III)] Dichloride Permanganate and [Hexaammincobalt(III)] dibromide Permanganate, $[Co(NH_3)_6]X_2(MnO_4)$, X = Cl, Br

The [hexaamminecobalt(III)] dichloride permanganate and dibromide permanganate $[Co(NH_3)_6]X_2(MnO_4)$ (X = Cl, Br) were prepared by a direct combination of one equiv. of powdered $[Co(NH_3)_6](MnO_4)_3$ and eight equiv. of $[Co(NH_3)_6]X_3$ (X = Cl, Br) in a small amount of hot water. On cooling, small strips of black crystals are deposited, which are washed with a sufficient quantity of water and dried at 50 °C [121]. The chloride compound was prepared at room temperature, which resulted in the same product in 16.4% yield due to the solubility of [hexaammincobalt(III)] dichloride permanganate in water at room temperature (7.89 g/100 mL) [61]. These were also prepared directly from [hexaamminecobalt(III)] salts with 1.5 equivalent of potassium permanganate solutions keeping the solution for 6 h [121].

The chloride salt is not very stable and decomposes in its aq. solution, releasing chloride ions completely, but it dissolves well in luteocobaltic chloride solution without decomposition. The bromide salt could be dried at 50 °C without decomposition. It has shiny hexagonal blades, similar to the chloride analog, but the bromide compound is much more stable. Water does not seem to split even when boiling. The chloride salt is insoluble in aliphatic and aromatic hydrocarbons, acetone, and chlorinated solvents such as $CCl_4$, chloroform, or dichloromethane, but it is soluble in DMF (0.848 g/100 mL) and decomposes in DMSO immediately. It also decomposes in a wet state in a day, but when it is dry and in the absence of light, it can be stored for several days [59]. Both compounds detonate on heating, releasing ammonia, but they are not sensitive to the impact of a hammer [121].

The chloride salt forms red platelets, monoclinic, space group $P2_1/c$, $a$ = 13.6133 Å, $b$ = 7.3658 Å, $c$ = 12.3682 Å; $\beta$ = 108.547°, Z = 4, $D_{calcd.}$ = 1.983 g/mL, T = 163 K, V = 1175.78 $Å^3$ [61]. The elementary cell contains four formula units, whereas the asymmetric unit contains two halves of the complex cation, two chloride ions, and one permanganate ion. There are two differently distorted octahedral cations (labeled as A and B), which bonded to the permanganate oxygens with hydrogen bonds with different lengths. No direct metal–metal interactions were found (Figure 5).

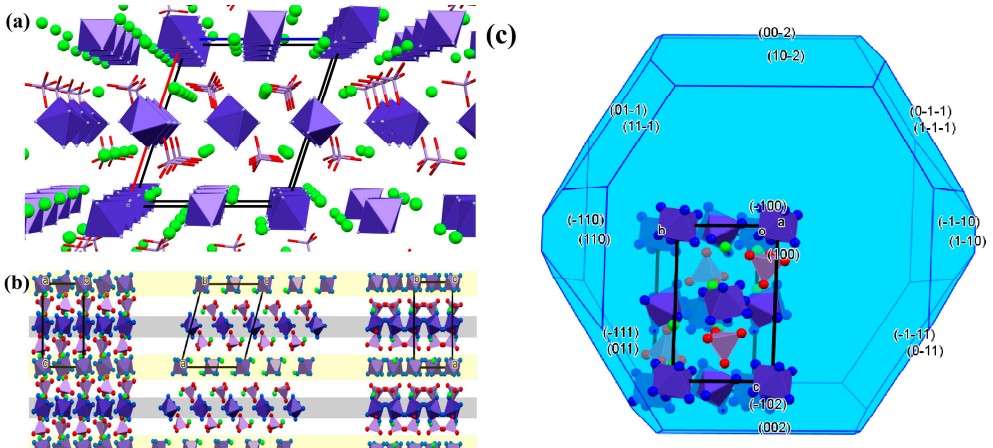

**Figure 5. (a)** Packing arrangement in the crystal of hexaamminecobalt (III) dichloro permanganate. **(b)** Packing arrangement in the crystal of [hexaamminecobalt(III)] dichloride from various directions. Different cation layers are indicated by colored rectangles. Layers of cation A are indicated by yellow, and layers of cation B are indicated by gray (the structure is drawn in stick representation). **(c)** The BDFH–predicted morphology of [hexaamminecobalt(III)] dichloride. Reproduced from [61].

There are two cationic layers in the structure; cation A is placed together with Cl anions, whereas cation B is placed in a different kind of layer without any chloride ions. The second type of chloride ion is pushed into the anionic layer formed by the permanganate ions. The permanganate ion layers are in close contact with the second (B) cation layers, and they are further from the first (A) cationic layers in which the chloride ions shadow the positive charges. A total number of 25 and mostly strong hydrogen bonds occur between the complex cations and the two types of anions. Each ammonia has 3–5 hydrogen bonds with the anions. The chlorides involve 5 and 6 hydrogen bonds, which are weaker than that made by permanganate ions. The intermolecular interactions of the two crystallographically independent cations were compared by a Hirshfeld surface analysis, and 2D fingerprint plots were generated and shown in Figure 6.

The fingerprint plots for the two cations show marked differences. The strongest H–bonds are formed with cation B (with a permanganate ion), which is indicated by the much lower $d_e$ and $d_i$ values. The spike of the N–H⋯Cl interaction for cation B appears at higher de values, and the spike for the N–H⋯O interactions for cation A is missing. It shows that the interactions between the cation A and the permanganate ions are slightly loose, whereas for the cation B, the N–H⋯O spike is pronounced. The number of the N–H⋯Cl interactions of cation A is higher than that for cation B (indicated by red) [61].

The vibrational spectra (IR and Raman) and factor group analysis were carried out for the chloride salt, and all the normal modes for the cation and anion have been assigned. A relative bond strength parameter ($\varepsilon$) for the ammonia molecules in ammine complexes [117], as defined by Grinberg, was found to be 0.94 and 0.90 for the coordinated ammonia, which shows that the average strength of the hydrogen bonds in this compound is between the strength of an average hydrogen bond strengths in $[Co(NH_3)_6]Cl_3$ [120] and $[Co(NH_3)_6](MnO_4)_3$ [61,117]. Its solid phase UV–Vis spectrum at room temperature consists of strongly overlapping bands of four possible d–d transitions of the $[Co(NH_3)_6]^{3+}$ cation and CT bands of the permanganate anion. The $^1A_1 \rightarrow {}^1T_1$ and $^1A_1 \rightarrow {}^1T_2$ transitions of the octahedral $Co^{III}$ cation are spin-allowed. The distortion due to hydrogen bonds results in trigonal distortion (compression) [61].

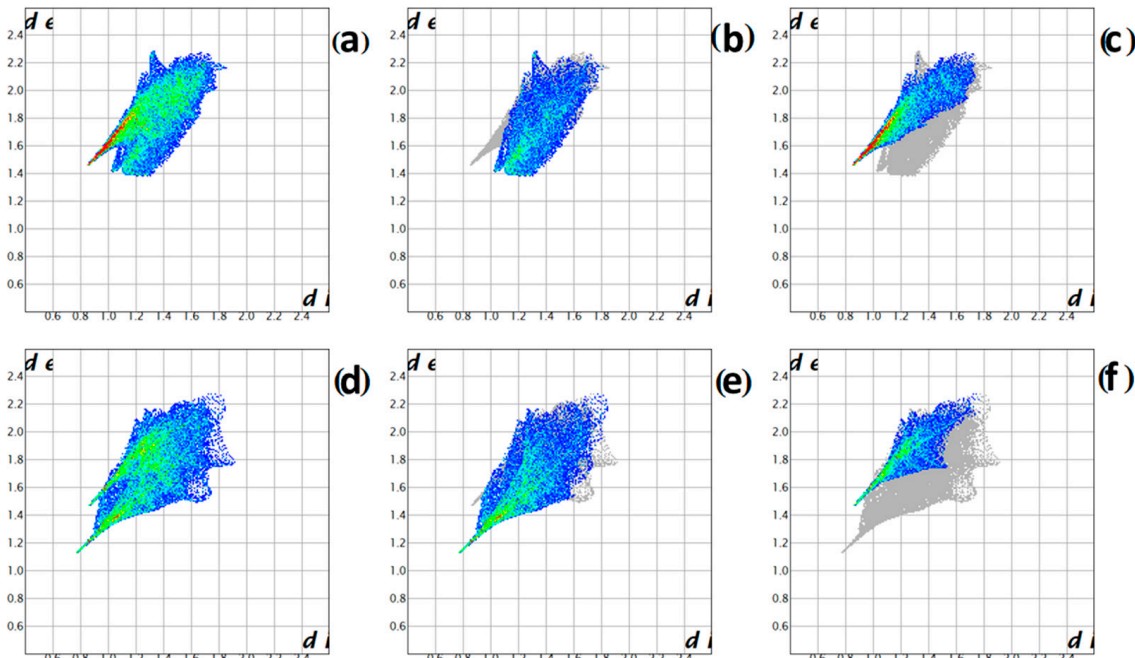

**Figure 6.** Hirshfeld surface analysis and 2D fingerprint plots of [hexaamminecobalt(III)] dichloride permanganate. Reproduced from [61]. (**a**) intermolecular interactions of cation A, (**b**) N–H⋯O interactions of cation A, (**c**) N–H⋯Cl interactions of cation A, (**d**) intermolecular interactions of cation B, (**e**) N–H⋯O interactions of cation B, and (**f**) N–H⋯Cl interactions of cation B.

The chloride salt is explosive upon heating, but a slow heating rate (2 °C/min) resulted in stepwise decomposition reactions. The first decomposition step was observed at 107 and 129 °C in inert and air atmospheres, respectively. The DSC peak temperatures were the same in $O_2$ and $N_2$ atmospheres (109 °C). The second decomposition step was observed at 134 °C in both atmospheres. The reaction heat of each step (−107.1 and −260.8 kJ/mol in the first and −90.3 and −64.5 kJ/mol in the second decomposition step, in $O_2$ and $N_2$ atmospheres, respectively) was found to be different. Although the outer oxygen does not play a direct role in starting the decomposition reaction, an indirect influence was found in the reaction heat via the formation of endothermic nitrogen oxides, such as NO or $N_2O$, detected by TG-MS [61]. Two other decomposition steps were observed at 382 and 441 °C under $N_2$ [61], whereas, in air, the oxidizable residues were completely oxidized and disappeared at 378 °C [61]. TG–MS measurements show that the first three decomposition steps consist of redox reactions because $H_2O$, NO, and $N_2O$ redox products were formed. The first redox reaction is a reaction between the ammonia ligands and permanganate ions. Water and $N_2$ formed in all three, whereas $N_2O$ only in the first and third, and NO in the second and third steps. Ammonia is also evolved because it is not enough permanganate to oxidize all ammonia molecules. There was no oxygen evolution in any step.

With the use of toluene as a heat-absorbing medium to avoid local overheating due to the exothermicity of the first decomposition step, an isotherm-controlled temperature decomposition was performed at 110 °C. The reaction temperature could not be exceeded the boiling point (110 °C) of toluene until liquid toluene is present. The $Co^{III}$ centers and permanganate ion are oxidants, whereas the chloride ions and ammonia act as reducing agents. The amorphous decomposition product was leached with water, and the aqueous extract contained $[Co(NH_3)_6]Cl_3$, $[Co(NH_3)_6]Cl_2$, $NH_4NO_3$, and $NH_4Cl$, whereas the water-insoluble product was the same todorokite-like phase as in the case of $[Co(NH_3)_5Cl](MnO_4)_2$ [61], $(NH_4)_4Co_2Mn_6O_{12}$. Based on magnetic measurements, cobalt is in the trivalent high-spin state, whereas the manganese is in the divalent or trivalent

high-spin state; thus, the todorokite-like compound probably contains $Mn^{II}_4Mn^{III}_2O_{12}^{10-}$ manganese oxide network with square-shaped channels [61].

Heating of the decomposition products in toluene with (removal of a part of Co in soluble compounds form) and without (keeping the Co:Mn = 1:1 ratio) aqueous leaching at 500 °C resulted in Co-Mn spinels with cubic and tetragonal structures and various Co:Mn ratios depending on the reaction conditions (Scheme 3) [61].

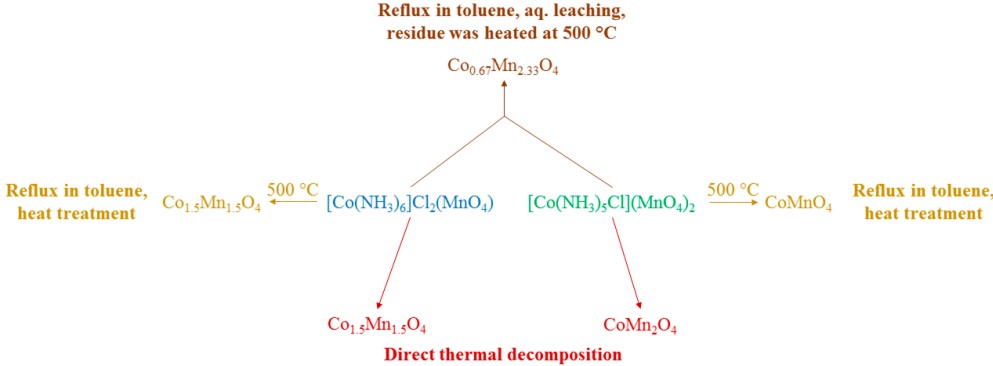

**Scheme 3.** The thermal decomposition of $[Co(NH_3)_5Cl](MnO_4)_2$. Reproduced from [59].

The heat treatment of the solid phase decomposition residue at 500 °C resulted in cubic spinel with $Co_{1.5}Mn_{1.5}O_4$ ($MnCo_2O_4$ type) composition. The toluene-derived material without aqueous leaching resulted in tetragonal $Co_{1.5}Mn_{1.5}O_4$, whereas after aq. leaching, a $Co_{0.67}Mn_{2.33}O_4$ phase was obtained [61]. The photocatalytic activity of these spinel-like phases was observed in the degradation of harmful organic dyes such as Congo red depending on their composition and synthesis conditions [61].

### 2.6.6. Potassium [Hexaamminecobalt(III)] Dichloride Dipermanganate

Klobb analyzed the hexagonal crystalline by-product formed during the synthesis of [hexaamminecobalt(III)] permanganate and found that the compound contains chloride and potassium as well. This compound is soluble in water and crystallizes slowly at high concentrations only. It forms pretty crystals with very clear contours in a beautiful violet color. This salt can be prepared in a pure state by mixing a concentrated cold solution of 1 equivalent of [hexaamminecobalt(III)] chloride with 1.5 equivalent of potassium permanganate. The purple salt crystallizes in several hours [123].

The double salt can also be prepared with a dissolution of [hexaamminecobalt(III)] dichloro permanganate in a concentrated solution of KCl, together with [hexaamminecobalt(III)] chloride as a by-product:

$$2Co(NH_3)_6Cl_2MnO_4 + KCl = Co(NH_3)_6Cl_3 + K[Co(NH_3)_6Cl_2(MnO_4)_2]$$

[Hexaamminecobalt(III)] permanganate also reacts with potassium chloride:

$$Co(NH_3)_6(MnO_4)_3 + 2KCl = KMnO_4 + K[Co(NH_3)_6Cl_2(MnO_4)_2]$$

The double salt forms small violet or black crystals, according to the thickness, often with a greasy luster. It is very well soluble in water with partial decomposition. Evaporation of its aq. solutions in cold leaves back a residue in which KCl, $[Co(NH_3)_6](MnO_4)_3$, and $[Co(NH_3)_6]Cl_3$ could be identified. It is insensitive to the impact of a hammer but detonates on heating, releasing ammonia [123]. Klobb declares that similar reactions of KBr and $NH_4Cl$ resulted in hexagonal blades under analog conditions, which were supposed to be the analog bromide or ammonium compounds, and analysis has not been performed [123].

## 3. Conclusions

The available data about permanganate, perrhenate, and pertechnetate salts of transition metal ammonia complexes, including the mono-, di-, tri-, tetra-, penta- and hexaammine derivatives, have been comprehensively reviewed.

The synthetic procedures to prepare the ammonia complexes of transition metal permanganate, pertechnetate, and perrhenate (VIIB group tetraoxometallates) salts have been given and compared. The available data about the structure and spectroscopic properties of these compounds, including the presence of hydrogen bonds that act as redox reaction centers under their thermal decomposition, have been presented and evaluated in detail. The nature of the thermal decomposition products has also been summarized. The available pieces of information about the role of the ammine complexes of the transition metal permanganate salts in organic oxidation reactions, like the oxidation of benzyl alcohols and regeneration of oxo-compounds from oximes and phenylhydrazones including the kinetics of these processes, have also been collected. The physical and chemical properties, including the thermal decomposition characteristics of the known diammine (Ag(I), Cd, Zn, Cu(II), Ni(II)), triammine (Ag(I)), and simple or mixed ligand tetraammine (Cu(II), Zn, Cd, Ni(II), Co(II), Pt(II), Pd(II), Co(III)), Ru(III), pentaammine (Co(III), Cr(III), Rh(III) and Ir(III)) and hexaammine (Ni(II), Co(III), Cr(III)) complexes of transition metals with tetraoxometallate(VII) anions (M = Mn, Tc and Re) have been summarized.

The preparation of properties of some mixed ligand (e.g., $[Ru(NH_3)_4(NO)(H_2O)](ReO_4)_2$ or $[Co(NH_3)_5(H_2O)](ReO_4)_2$) or mixed anionic complexes, with mixed anions in the inner (e.g., $[Co(NH_3)_5X](MnO_4)_2$ (X = Cl, Br)) and outer (e.g., $[Co(NH_3)_6]Cl_2(MnO_4)$) coordination sphere have been discussed. The properties of special complexes such as inner sphere tetraoxometallate coordinated species like $[Co(NH_3)_5ReO_4]X_2$ (X = Cl, NO_3, ClO_4, ReO_4) and mixed cationic/anionic ones as $K[Co(NH_3)_6]Cl_2(MnO_4)_2$ have been included.

**Funding:** This research received no external funding.

**Conflicts of Interest:** The author declares no conflict of interest.

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
