# Peer review of "Review on the Chemistry of [M(NH3)n](XO4)m (M = Transition Metal, X = Mn, Tc or Re, n = 1–6, m = 1–3) Ammine Complexes"

_inorganics, doi:10.3390/inorganics11070308_

Round 1

Reviewer 1 Report

  This paper is a review of [M(NH3)n](XO4) (Mn, Tc, Re), which could be valuable because there have been limited reviews on this topic.

However, similar descriptions are repeatedly stated, which may be important for the properties of each compound, but the descriptions are too detailed and give a feeling of redundancy. This paper should be more compactly organized and presented in an easy-to-read format by using more figures and tables and using detailed descriptions as supplementary data. Further points to be reconsidered and corrected are listed below:

1) Abstract does not effectively represent the content of the text.

2) In Scheme 1 and 3, the numbering of the complexes should be removed or corrected.

3) Tables 1, 2, and 3 are not used for main discussions. It would be better that they are included as supplementary information.  Instead, tables summarizing bond distances and angles may be important for the discussion in the text.

4) Is reference 121 correct?

5) In Line 1156, [Co(NH3)5ReO3](NO3)2 should be listed as [Co(NH3)5OReO3](NO3)2.

6) Figure 7 is not cited in the text. Note that Figure 7 is a mistake for Figure 5. In the caption of this figure, (a), (b), and (c) are not explained. In addition, the figures of labled A and B described in Line 1395 should be included in Figure 5.

7) Figure 5 is an error for Figure 6.

8) In Conclusion, more detailed descriptions are needed to explain what the author wanted to say in this review.

Author Response

First of all, I would like to express my many thanks for the reviewer efforts to improve my manuscript’s quality and the valuable suggestions, which were followed. Detailed answers are given after the suggestions.

Reviewer 1.

This paper is a review of [M(NH3)n](XO4) (Mn, Tc, Re), which could be valuable because there have been limited reviews on this topic.

However, similar descriptions are repeatedly stated, which may be important for the properties of each compound, but the descriptions are too detailed and give a feeling of redundancy. This paper should be more compactly organized and presented in an easy-to-read format by using more figures and tables and using detailed descriptions as supplementary data. Further points to be reconsidered and corrected are listed below:

I tried to remove the similar (repetitive) descriptions from the text. The original raw manuscript contained more Figures, but the copyright costs increased so much that I had to decrease the number of Figures used in the submitted manuscript.  

  • Abstract does not effectively represent the content of the text.

The abstract has been changed and supplied with more pieces of information.

  • In Scheme 1 and 3, the numbering of the complexes should be removed or corrected.

These have been corrected.

  • Tables 1, 2, and 3 are not used for main discussions. It would be better that they are included as supplementary information. Instead, tables summarizing bond distances and angles may be important for the discussion in the text.

The discussion about the Tables is inserted.  The large number of bonds (direct coordinative and hydrogen bonds) in the compounds would result in too many Tables. Therefore, we always gave the references, where these data are available. However, some important bond distances/angles are discussed in the text written about the particular compound.

  • Is reference 121 correct?

The reference has been corrected.

  • In Line 1156, [Co(NH3)5ReO3](NO3)2 should be listed as [Co(NH3)5OReO3](NO3)2.

The formula has been corrected.

  • Figure 7 is not cited in the text. Note that Figure 7 is a mistake for Figure 5. In the caption of this figure, (a), (b), and (c) are not explained. In addition, the figures of labled A and B described in Line 1395 should be included in Figure 5.

The captions have been changed, and the mistakes were removed.

  • Figure 5 is an error for Figure 6.

The numbering has been revised.

8) In Conclusion, more detailed descriptions are needed to explain what the author wanted to say in this review.

                The conclusion has been revised.

Reviewer 2 Report

This review provides exhaustive information on the chemistry of ammonia complexes of transition metals with tetraoxametalate anions of Group 7 metals. It is difficult to assess the relevance of such a review, but certainly its publication is very useful, since it combines information scattered in many sources of different times and different accessibility. In this regard, I would strongly recommend that the author give as much as possible electronic links to the sources used, including DOI and others, for example, https://www.researchgate.net/publication/257922081_ChemInform_Abstract_Beliefs_and_Facts_in_Permanganate_Chemistry_-_An_Overview_on_the_Synthesis_and_the_Reactivity_of_Simple_and_Complex_Permanganates

 for Reference 65. The references should also indicate the name of the series for such journals as Acta Cryst. and Z. Naturforsch. References 87 and 88 are the same.

In general, in my opinion, the proposed review sins with unnecessary details in many places. The reasons for including Figures 2 and 3 are not very clear, while the summary tables make a good impression.

It is also necessary to carefully proofread the text to correct various typographical errors, including missing parentheses and letters in subheading titles (eg. 2.6.2 and 1.6.5).

Author Response

First of all, I would like to express my many thanks for the reviewer’ efforts to improve my manuscript’s quality and the valuable suggestions, which were followed. Detailed answers are given after the suggestions.

Reviewer 2.

This review provides exhaustive information on the chemistry of ammonia complexes of transition metals with tetraoxametalate anions of Group 7 metals. It is difficult to assess the relevance of such a review, but certainly its publication is very useful, since it combines information scattered in many sources of different times and different accessibility. In this regard, I would strongly recommend that the author give as much as possible electronic links to the sources used, including DOI and others, for example, https://www.researchgate.net/publication/257922081_ChemInform_Abstract_Beliefs_and_Facts_in_Permanganate_Chemistry_-_An_Overview_on_the_Synthesis_and_the_Reactivity_of_Simple_and_Complex_Permanganates

The links (Crossref, Google Scholar) will be given by MDPI after accepting the paper.

For Reference 65. The references should also indicate the name of the series for such journals as Acta Cryst. and Z. Naturforsch. References 87 and 88 are the same.

Ref. 65 is a journal without a series. The other references have been revised.

In general, in my opinion, the proposed review sins with unnecessary details in many places. The reasons for including Figures 2 and 3 are not very clear, while the summary tables make a good impression.

The text has been shortened and some parts were condensed. Figures 2 and 3 explaining has been improved. Some of the detailed information, however, is given due to a major part of non-easily available sources in the reference list.

It is also necessary to carefully proofread the text to correct various typographical errors, including missing parentheses and letters in subheading titles (eg. 2.6.2 and 1.6.5).

I checked all the text and revised these and similar mistakes.

Round 2

Reviewer 1 Report

Because the manuscript has been corrected, this paper could be ready for publication. However, there are still a few points that need to be corrected. A list is provided below:

(1)  Line 162: “[Ag(NH3)2MnO4” should be “[Ag(NH3)2]MnO4”.

(2)  Line 254: “([Ag(NH3)2]MnO4” should be “([Ag(NH3)2]MnO4)”.

(3)  Line 305: “[Zn(ReO4)2” should be “[Zn(ReO4)2]”.

(4)  Lines 371, 372: L = 4.8.10-20 for Cu(OH); “Ksp” could be more usual than “L” for solubility product. Further, L = 7.81.10-30 for [Cu(NH3)4](MnO4)2; the value is not found in the refs. 73 and 76.

(5)  Line 750: “M[NH3]4(ReO4)2” should be “[M(NH3)4](ReO4)2”.

(6)  Line 835: µeff = 4.50-4.54 should be added by unit (B.M.).

(7)  Line: ref. 121 was not corrected. Ref. 121 describes the Ag complex, not for Co complex.

(8)  In the caption of Figure 6, (a) ~ (f) should explained.

Author Response

Thank you for your careful work in checking and improving my manuscript. 

All the mistakes have been revsied. 

In detail: 

(1)  Line 162: “[Ag(NH3)2MnO4” should be “[Ag(NH3)2]MnO4”.

It has been revised.

(2)  Line 254: “([Ag(NH3)2]MnO4” should be “([Ag(NH3)2]MnO4)”.

It has been revised.

(3)  Line 305: “[Zn(ReO4)2” should be “[Zn(ReO4)2]”.

It has been revised.

(4)  Lines 371, 372: L = 4.8.10-20 for Cu(OH); “Ksp” could be more usual than “L” for solubility product. Further, L = 7.81.10-30 for [Cu(NH3)4](MnO4)2; the value is not found in the refs. 73 and 76.

These have been revised. It is really in ref. 72.

(5)  Line 750: “M[NH3]4(ReO4)2” should be “[M(NH3)4](ReO4)2”.

It has been revised.

(6)  Line 835: µeff = 4.50-4.54 should be added by unit (B.M.).

It has been revised.

(7)  Line: ref. 121 was not corrected. Ref. 121 describes the Ag complex, not for Co complex.

It has been revied (ref. 60 is the correct one).

(8)  In the caption of Figure 6, (a) ~ (f) should explained.

Yes, it has been revised.

Reviewer 2 Report

I fully agree with the author that the need for certain details in the presented review depends on the availability of the cited sources for readers (some have DOI and are easily accessible, others can be found in open electronic archives, and some are simply not available in electronic form and in many libraries. Unfortunately, this is very difficult to verify, since the author did not respond to my remark and took the trouble to bring the list of references into a form convenient for work.

Author Response

Thank you for your careful work in behalf of my manuscript quality improving.

 I am sorry, I misunderstood the reviewer’s former request (to see by himself/herself the papers which have no DOI). I believed that the respected reviewer wanted I gave the DOI for papers according to the journal rules (it is done by the journal type-editors).

DOI or electronic versions could not find for the next papers: I used printed versions from different libraries (ca. 30 papers). Reference numbers of 1, 39, 44-47, 51-55, 57, 66, 68, 71, 73, 77, 84-86, 88, 89, 99, 104, 109, 111-113, 115, 118, 120, 127.  Furthermore, there is DOI for a paper, sometimes that is a problem to reach due to paying conditions.